# Protein design of two-component tubular assemblies similar to cytoskeletons

Masahiro Noji [1,2,3], Yukihiko Sugita [4,5,6], Yosuke Yamazaki [7,8], Makito Miyazaki [6,7,8,9] & Yuta Suzuki [3,6,9] ✉

Recent advances in protein design have ushered in an era of constructing intricate higher-order structures. Nonetheless, orchestrating the assembly of diverse protein units into cohesive artificial structures akin to biological assembly systems, especially in tubular forms, remains elusive. To this end, we develop a methodology inspired by nature, which utilises two distinct protein units to create unique tubular structures under carefully designed conditions. These structures demonstrate dynamic flexibility similar to that of actin filaments, with cryo electron microscopy revealing diverse morphologies, like microtubules. By mimicking actin filaments, helical conformations are incorporated into tubular assemblies, thereby enriching their structural diversity. Notably, these assemblies can be reversibly disassembled and reassembled in response to environmental stimuli, including changes in salt concentration and temperature, mirroring the dynamic behaviour of natural systems. This methodology combines rational protein design with biophysical insights, leading to the creation of biomimetic, adaptable, and reversible higher-order assemblies. This approach deepens our understanding of protein assembly design and complex biological structures. Concurrently, it broadens the horizons of synthetic biology and material science, holding significant implications for unravelling life's fundamental processes and enabling future applications.

Life phenomena rely on the dynamic and reversible assembly and disassembly of various higher-order protein assemblies. Actin filaments[1,2] and microtubules[3,4] in the cytoskeleton and the capsid proteins of viruses[5,6] are examples of such naturally occurring structures. These are tightly regulated in function and complexity. Synthesising higher-order structures of heterogeneous protein units poses a significant challenge, particularly regarding replicating the diversity and flexibility inherent to natural assemblies. Although recent advances in computational design have enabled the creation of artificial higher-order protein structures from two protein components[7–9],

the design of heterogeneous higher-order protein assemblies with the flexibility and reversible assembly/disassembly characteristics of natural structures, especially tube structures, remains a formidable challenge.

Herein, we introduced a methodology inspired by nature that draws inspiration from the principles underlying natural protein complexes. By integrating rational protein design with biophysical insights to optimise assembly conditions, this approach recapitulates the flexibility and reversible assembly principles of natural systems. Using this methodology, we created an assembly of two distinct

[1]Research Fellow of Japan Society for the Promotion of Science, Tokyo, Japan. [2]Graduate School of Human and Environmental Studies, Kyoto University, Kyoto, Japan. [3]Institute for Integrated Cell-Material Sciences, Kyoto University, Kyoto, Japan. [4]Institute for Life and Medical Sciences, Kyoto University, Kyoto, Japan. [5]Graduate School of Biostudies, Kyoto University, Kyoto, Japan. [6]Hakubi Center for Advanced Research, Kyoto University, Kyoto, Japan. [7]Graduate School of Science, Kyoto University, Kyoto, Japan. [8]RIKEN Center for Biosystems Dynamics Research, Yokohama, Japan. [9]PRESTO, JST, Saitama, Japan. ✉e-mail: suzuki.yuta.2m@kyoto-u.ac.jp

protein components, successfully forming unique two-component tube structures. This development represents a significant step toward replicating the properties of complex natural structures at the molecular level.

## Results

### The concept of protein assembly design

The concept of this approach draws inspiration from natural biological systems, employing rational design principles to integrate naturally occurring 'heterolinkers' with 'scaffold proteins' to streamline the design process. For the heterolinker, we chose the heterodimeric peptide pair 'MBD3L2 (M3L2)/p66α' (Fig. 1a). Our choice of M3L2/p66α was influenced by its role in the M3L2-NuRD complex, where the heterodimeric 'M3L2/p66α' anti-parallel coiled-coil domain is essential for complex assembly[10]. Given the moderate denaturation midpoint temperature ($T_m$) of M3L2/p66α ($T_m = 35\,°C$)[11], we anticipated that M3L2/p66α would provide a balance between stability and reversible assembly control through temperature modulation. We then sought to identify a scaffold protein that could connect the heterolinker in the simplest manner possible. The positions of connecting sites at the corners of such scaffold proteins facilitate the desired assembly formation[12]. Therefore, we chose the '*Pseudomonas fluorescens* PuuE allantoinase (PuuE)', a homotetramer with $C_4$ symmetry where each C-terminus is located at each vertex of the quaternary structure (Fig. 1b)[13]. This arrangement enabled straightforward genetic fusion of heterolinkers to the scaffold's C-termini, leveraging the specificity and reversibility of heterolinker interactions to drive assembly formation. This approach simplifies the assembly process and enhances

expression and purification efficiency for each protein unit, preventing spontaneous assembly and ensuring the controlled formation of higher-order structures.

We constructed protein units 'PuuE-M' and 'PuuE-p' through genetic engineering, fusing M3L2 and p66α to the C-terminus of PuuE, respectively (Fig. 1c). AlphaFold2 (AF2)[14,15] modelling suggested a configuration with a relatively flexible orientation of M3L2 in PuuE-M, whereas a highly constrained orientation of p66α in PuuE-p (Fig. 1c and Supplementary Fig. 1). Owing to the constrained orientation of PuuE-p, an angular interface was formed between PuuE-M and PuuE-p, facilitating the formation of a 'closed' structure (i.e., ring, cage, or tube) rather than an 'open' structure (i.e., sheet) (Fig. 1d). In addition, variations in the number of PuuE-M and PuuE-p units were predicted to result in a diverse range of closed structures. Protein expression in *Escherichia coli* provided both constructs in a soluble form, facilitating their purification. In isolation, neither protein unit exhibits self-assembly (Supplementary Fig. 2a, b). However, when combined under optimised conditions (discussed in the following section), we successfully observed the expected closed structure, chessboard-patterned tube (PuuE tube), using negative-stain transmission electron microscopy (nsTEM) (Fig. 1e and Supplementary Fig. 2c). Although previous studies have assembled cages[7,8], sheets[9], and three-dimensional (3D) crystals[16] using two-component protein systems, this study is unique in that tube structures were successfully created. While tubular structures have been designed using computational methods[17,18], these have been achieved in single-component systems. To the best of our knowledge, this study is the first successful creation of tubular structures composed of two different components.

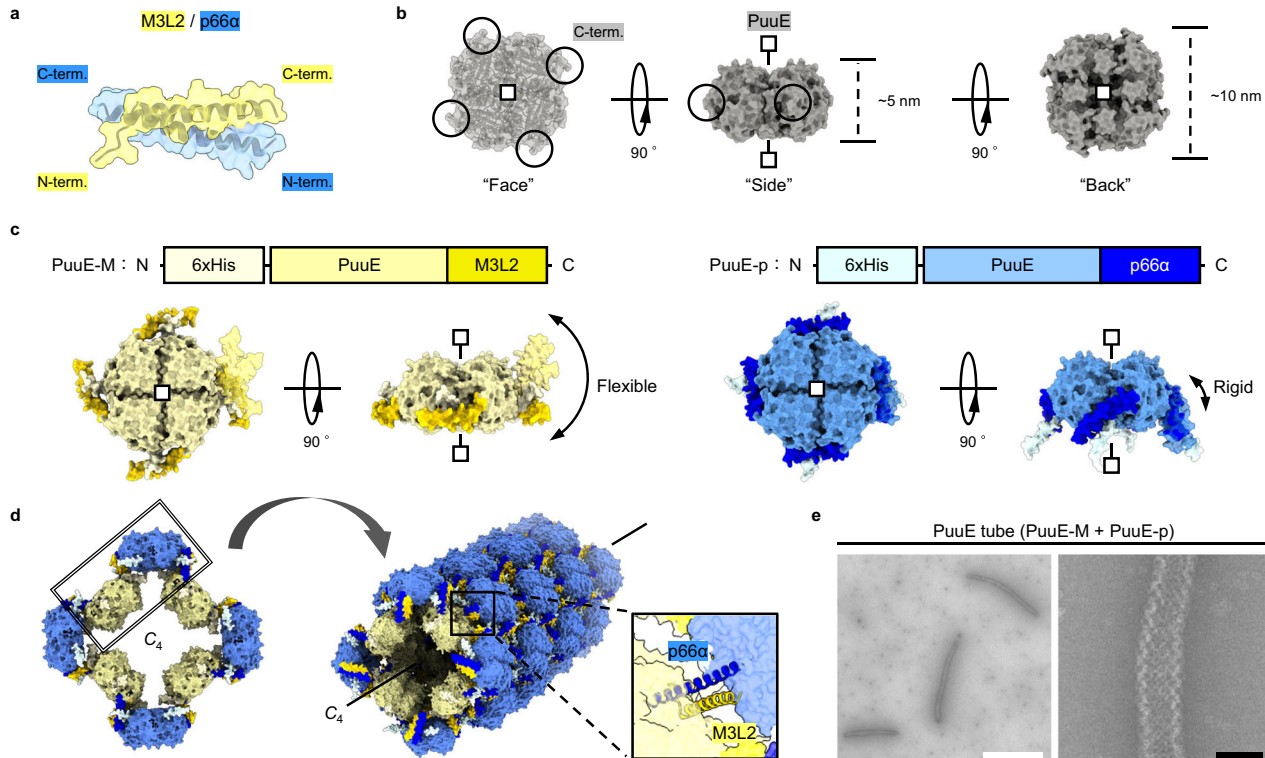

**Fig. 1 | Construction of PuuE tube. a** AF2 prediction of the heterodimeric peptide pair, M3L2 (yellow) and p66α (blue). **b** Crystal structure of PuuE (PDB ID: 3CL6). C-terminus positions are circled. Detailed structure, face, side, and back are shown for clarity. **c** Schematic diagram of the protein sequence (top) and the AF2-predicted structures of PuuE-M and PuuE-p (bottom). PuuE-M and PuuE-p are coloured yellow and blue to match the respective peptides and overall structure to clear the tube structure (**d**). Symmetry of protein is represented with lines and square markers. The peptide parts, M3L2 and p66α, are highlighted in darker colours. The flexibility of the connecting segments, M3L2 and p66α, as predicted by AF2, is demonstrated using a single peptide structure shown on the right. Further details are provided in Supplementary Fig. 1. **d** Predicted model of closed structure, e.g., tube structure with $C_4$ symmetry, consisting of PuuE-M and PuuE-p. The repeat unit was highlighted within a double-framed box. **e** nsTEM images of tubular assemblies constructed from PuuE-M and PuuE-p; 12.5 μM PuuE-M and 12.5 μM PuuE-p in NaCl (+) buffer was incubated at 40 °C for 24 h and imaged via nsTEM. Scale bars, 1 μm (white), 50 nm (black).

## The condition design of tubular assemblies

Based on established principles observed in biological systems, including actin filaments[19,20], microtubules[21,22], and amyloid fibrils[23,24], protein concentration, temperature, time, and salinity have a significant influence on assembly formation. Thus, we carefully tailored assembly conditions to exploit the complex interactions between these factors. This approach allowed us to optimise experimental conditions for constructing the desired tubular structures.

First, we focused on the dependency of PuuE tube assembly on protein concentration (Supplementary Fig. 2c). Mixing PuuE-M and PuuE-p at a concentration of 250 nM each (considering tetramer equivalence) led to the formation of tubular structures after an incubation period of 24 h at 40 °C, consistent with the dissociation constant ($K_d$) for M3L2/p66α dimer formation, which is approximately 268 nM[11]. Increasing protein concentration to 2.5 μM markedly enhanced the quantity and length of formed tubular structures. Elevating the concentration to 12.5 μM for each component significantly increased tube formation efficiency, underscoring the concentration-dependent nature of PuuE-M- and PuuE-p-facilitated tubular assembly.

Next, PuuE tube formation kinetics were investigated. The incubation of mixtures containing 12.5 μM of each protein at 40 °C resulted in the formation of nascent tube structures within 30 min, evolving into distinguishable tubes spanning several hundred nanometres to 1 μm in length within 1–2 h (Fig. 2a, b and Supplementary Fig. 3a). Over time, these tubes elongated, reaching several micrometres in length after 24 h and extending up to ~5 μm after 48 h. In addition to the tubular assemblies, nsTEM observation also revealed the presence of ring-like structures under the same conditions (Supplementary Fig. 3b). These ring-like assemblies likely represent an early stage in the formation of tubular structures, as their geometry aligns with the design principle of a fixed angular connection at the p66α site of PuuE-p (Fig. 1c, d; and Supplementary Fig. 1). While the focus of this study was on the tubular structures, these observations provide further evidence for the versatility of the design in forming higher-order assemblies. Once formed, the tubes remained structurally stable for at least 1 month at 25 ± 1 °C (Supplementary Fig. 3c).

We then explored the influence of temperature on PuuE tube formation (Supplementary Fig. 4a). While the melting temperature of the M3L2/p66α dimer is around 35 °C, tube assembly was hardly observed at sufficiently lower temperatures of 20–25 °C, even after 24 h of incubation. Conversely, temperatures near $T_m$, specifically between 30 and 40 °C, markedly promoted tube formation. Therefore, temperatures below $T_m$ may excessively enhance the binding force between M3L2 and p66α, causing kinetic entrapment of assemblies. However, temperatures close to $T_m$ modulate this binding force, allowing the dynamic rearrangement of M3L2/p66α interactions under thermal fluctuations, thus facilitating the assembly of thermodynamically stable, ordered structures. This principle is consistent with general crystallisation theories[25,26] and reports on the formation of ordered structures in natural protein assemblies[23,24,27]. Importantly, temperatures above 45 °C led to thermal denaturation and aggregation of PuuE-M ($T_m = 46.2$ °C) and PuuE-p ($T_m = 48.1$ °C), significantly diminishing tube formation capabilities (Supplementary Fig. 4b, c). This finding implies that the original concept of tube formation with reversible temperature control was not realised.

## Reversibility of tubular assemblies

Finally, we examined the effects of salt concentration on PuuE tube assembly. We prepared mixtures with different NaCl concentrations ranging from 0 to 400 mM and incubated them at 40 °C for 24 h. Tube formation was clearly observed within the NaCl concentration window of 50–200 mM, with no tube formation detected outside this range (Fig. 2c, Supplementary Fig. 5b). Since both PuuE-M (pI = 6.44) and PuuE-p (pI = 6.02) were similarly charged under pH 8.0 (Supplementary Fig. 5a), tube formation at low salt concentrations was likely

inhibited by electrostatic repulsion. Conversely, moderate electrostatic shielding facilitated by 50–200 mM NaCl likely provided conducive conditions for tube assembly, whereas higher NaCl concentrations may have induced excessive shielding or aggregation due to salting out, inhibiting tube formation. This observation aligns with known phenomena in protein crystallisation, where electrostatic shielding above a certain threshold can prevent crystal growth[16,28–30], although crystals formed by a combination of electrostatic and

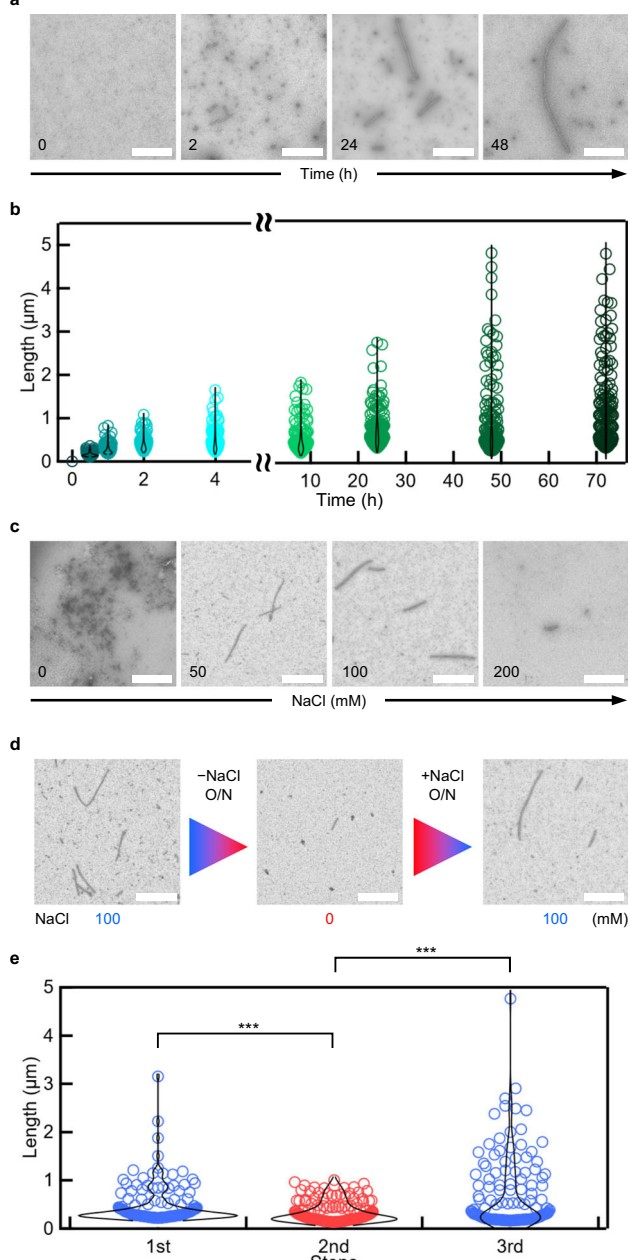

**Fig. 2 | Condition optimisation for PuuE tube assembly. a, b** The kinetics of tubular assembly. nsTEM images of tubular assembly (**a**) and length analysis (**b**). **c** nsTEM images of tubular assemblies with varying NaCl concentration. **d** nsTEM images showing the reversibility of tube structures with changing NaCl concentration. **e** Tube length analysis of nsTEM images. For tube length analysis, tubes were picked up and calculated from 5k images at each step; 150 tubes from the longest tube length were used at each data point. *** $p < 0.001$ ($p$-values; 1st–2nd = $9.98 \times 10^{-4}$ and 2nd–3rd = $3.63 \times 10^{-6}$, two-sided Welch's $t$ test). Scale bar, 1 μm. Source data are provided as a Source Data file.

hydrophobic interactions can remain stable up to approximately 200 mM NaCl[31]. The association of M3L2/p66α involves both electro-static and hydrophobic interactions[11], consistent with the latter scenario.

Given that tube formation was highly sensitive to NaCl con-centration, we considered the possibility that surface charge mod-ulation by affinity tags might influence assembly behaviour. To assess the impact of His-tags, we cleaved them from PuuE-M and PuuE-p using TEV protease and evaluated the assembly of the resulting proteins. These cleaved proteins formed chessboard-patterned tubes compar-able to those of the tagged counterparts (Supplementary Fig. 5c). However, efficient tube formation required a significantly higher NaCl concentration (400 mM). This increase likely resulted from reduced isoelectric points of PuuE-M (6.44 to 6.38) and PuuE-p (6.02 to 5.84) upon tag removal, necessitating stronger ionic shielding to counteract electrostatic repulsion. As previously noted, such high salt con-centrations also promoted nonspecific aggregation via salting-out effects, which competed with productive assembly. Based on these findings, we retained the His-tags in subsequent experiments to ensure consistent assembly under milder ionic conditions.

The salt-dependent PuuE tube formation and the dynamic nature of PuuE-M/PuuE-p interactions near their $T_m$ (35 °C) led us to hypo-thesise that tubes could undergo reversible disassembly and reas-sembly in response to changes in NaCl concentration. Confirming our hypothesis, tubes initially formed in 100 mM NaCl solution were sig-nificantly shortened when subjected to solvent exchange with 0 mM NaCl buffer (NaCl (−) buffer) and subsequent incubation at 40 °C for 24 h (Fig. 2d, e, Supplementary Fig. 5d). Subsequent solvent exchange with 100 mM NaCl buffer (NaCl (+) buffer) resulted in notable tube reassembly. This salt-concentration-driven reversibility, although divergent from the initial temperature-controlled reversibility hypothesis, marks a significant advance in artificial protein assembly design, allowing for the biomimetic replication of dynamic structural changes under relatively mild conditions, akin to the behaviour of actin filaments in cellular structures[19,20]. Unlike the irreversible aggre-gation observed in amyloid structures, our assemblies exhibit a reversible and dynamic assembly process akin to the cytoskeleton behaviour, successfully demonstrating the potential for the biomi-metic replication of natural cellular dynamics under controlled conditions.

## Diversity and flexibility of tubes

Based on these findings, we determined the ideal conditions for PuuE tube formation in 100 mM NaCl at 40 °C for 24 h. To further char-acterise the structural features of tubes formed under these condi-tions, cryo electron microscopy (cryo-EM) was employed (Supplementary Fig. 6, Supplementary Table 2). Analysis of 2D class-averaged images revealed a spectrum of tube diameters and sym-metries similar to the diversity observed in microtubules[32–35] (Fig. 3a). From these images, we successfully reconstructed the 3D structures with $C_4$, $C_5$, and $C_6$ symmetries within tube structures (Fig. 3b). Insights from the PuuE crystal structure[13], notably its unique central indentation on the back surface (Fig. 1b), allowed us to deduce that PuuE units are alternately oriented face-to-back across all 3D models. In addition, cryo-EM analysis suggested that connection flexibility allowed the contraction of the entire tube structure (Supplementary Fig. 6, Supplementary Movie 1). Although definitive conclusions are difficult owing to its inherent flexibility, the comparison of the cryo-EM 3D reconstruction with the AF2-predicted model of PuuE-p sug-gests that PuuE-p is less likely to fit inside the tube structure and instead fits better on the outside (Supplementary Movie 2). Fur-thermore, tubes with larger diameters, presumably having $C_7$ to $C_{10}$ symmetries, were identified at low resolution, likely owing to the flexibility of connection sites influencing tube structure. In fact, nsTEM and cryo-EM images frequently showed tubes appearing bent

or compressed (Supplementary Fig. 2–6). In contrast to prior stra-tegies by engineering on scaffold proteins itself to create higher-order protein assemblies[7–9,12,17,18,36], this approach integrates a flexible linker with the scaffold protein, resulting in varied structures and arrangements among higher-order assemblies. This variation in tube diameter, akin to that observed in microtubules[32–35], highlights the unique flexibility and adaptability of this method. Such structural diversity underscores the potential of this approach for creating functionally versatile protein assemblies. In addition, the flexibility of the linker resulting in various tube diameters is consistent with previous studies showing that flexible connections can result in dif-ferent structural assemblies[37,38]. Furthermore, the 3D maps sug-gested the presence of weak unintended interactions between PuuE units. However, as no higher-order assemblies containing tubular structures were observed when PuuE-M or PuuE-p were used alone, it is reasonable to conclude that the key interactions driving tube formation are designed interactions between the M3L2 and p66α components.

To further explore PuuE tube structure flexibility, we labelled tubes with Alexa Fluor 488 succinimidyl ester and observed them in real-time using total internal reflection fluorescence microscopy (TIRFM). Tube structures were constrained in the evanescent field by the depletion effect of methylcellulose contained in the obser-vation buffer and underwent thermally driven two-dimensional random bending (Fig. 3c, Supplementary Fig. 7a and Supplementary Movie 3). Analysis of the fluctuation in shape yielded the persistence length ($L_p$) of 19.7 μm (Fig. 3d and Supplementary Fig. 7b). $L_p$ is the mean length over which a semiflexible polymer remains straight, characterising polymer stiffness[39]. The $L_p$ value of the tube struc-tures is nearly equal to that of actin filaments measured in this study, 12.5 μm (Fig. 3d), and previously reported values of 9–20 μm[40]. Microtubules have much longer persistence lengths (0.1–10 mm)[41–43]. Conversely, intermediate filaments, another cytoskeletal fibre structure, typically have shorter persistence lengths (<1 μm)[44]. Although nanotubes longer than 15 μm would be required for a more accurate estimation of its persistence length, we can conclude that the current results suggest the nanotube is stiffer than intermediate filaments and more flexible than or as flexible as microtubules (Supplementary Fig. 7b).

## Emulation of actin filaments

Finally, we sought to modify the morphology of PuuE tube assemblies. Specifically, we hypothesised that grafting the D-loop of actin onto PuuE-M would produce tubes with a helical conformation reminiscent of actin filaments. The D-loop plays an important role in helical actin filament formation via hydrophobic pockets[45–48]. The hydrophobic nature of a prominent indentation on the 'back' side of PuuE (Sup-plementary Fig. 8a) guided our hypothesis. The loop structure on the back side of PuuE-M was chosen as the grafting site for the D-loop, and the 'PuuE(D-loop)-M' fusion construct was constructed (Fig. 4a). When PuuE(D-loop)-M was expressed in E. coli, it was found in the soluble fraction and was purified as PuuE-M. Since PuuE(D-loop)-M has a lower thermal stability ($T_m$ = 35.6 °C) than PuuE-M ($T_m$ = 46.2 °C, Supple-mentary Figs. 4b, 8b), we performed sample incubation at a lower temperature (30 °C). Although PuuE(D-loop)-M alone did not assem-ble, its combination with PuuE-p replicated the PuuE tube and intro-duced distinct helical patterns, with two and three tubes intertwined (PuuE D-loop tube), as verified via nsTEM (Fig. 4b and Supplementary Fig. 8c). The emergence of helical formations, absent in the PuuE-M and PuuE-p mixtures, clearly stems from D-loop integration. While the D-loop likely plays a crucial role in the helical formation of actin filaments[45–48], its complete mechanism remains unclear. Our study, by successfully grafting the D-loop to replicate actin-like helical struc-tures, provides additional insights into its role in filament architecture. This approach confirms the critical role of the D-loop in helical

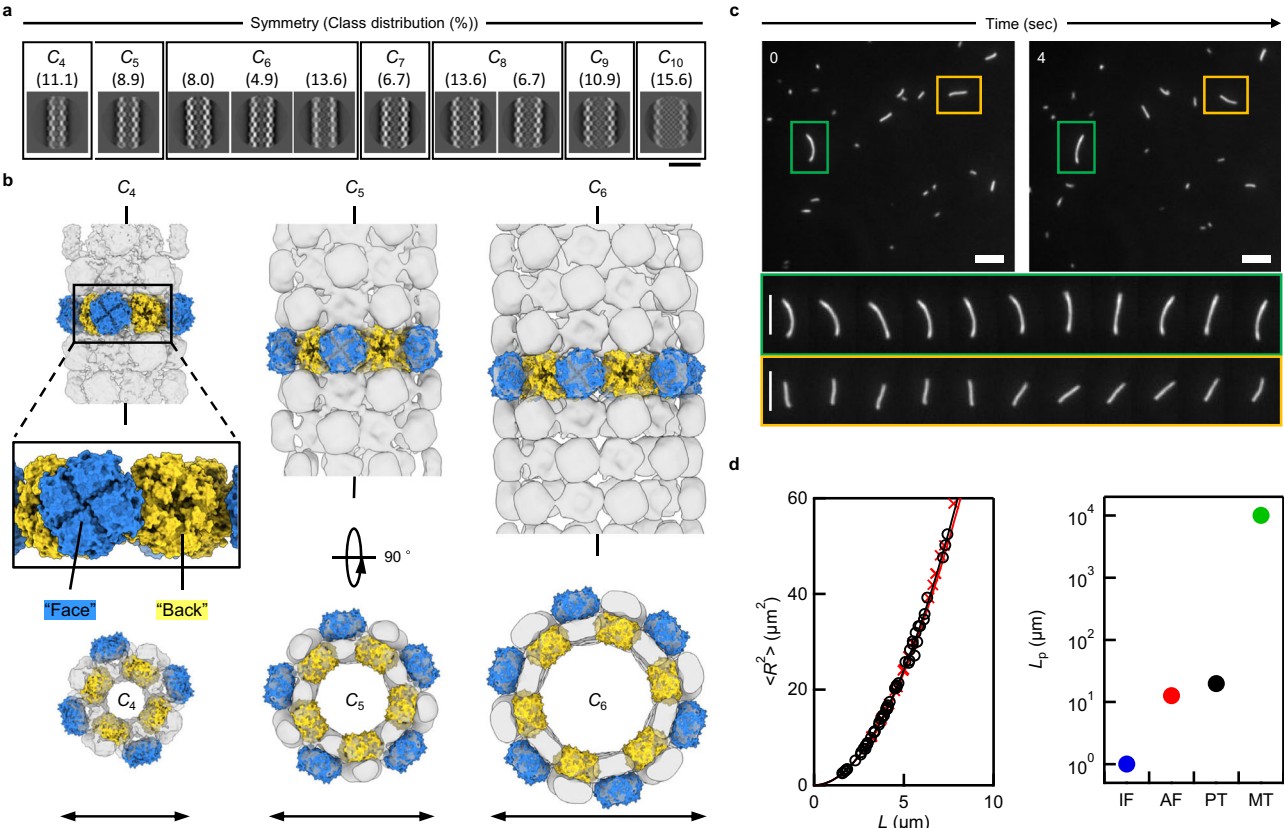

**Fig. 3 | Structural characterisation of PuuE tube. a** 2D class-averaged image of tube structures. The population of each structure was determined from the total pickings of 206,658 tube segments. Scale bar, 500 Å. **b** 3D reconstructed models of tube structures with $C_4$, $C_5$, and $C_6$ symmetries. The fitting results suggest that PuuE-p is less likely to fit into units located inside the tube structure and more likely to fit into units located on the outside. Based on the predictions, the units were colour-coded as shown in Fig. 1c. For visibility, only the molecular model of the PuuE (PDB ID: 3CL6) is overlaid on the 3D reconstructed model. **c** Time-lapse images of random bending of the tube structures monitored by TIRFM. Top: snapshots at the starting point (0 sec) and after 4 sec. Bottom: enlarged images of tubes in green or orange rectangles in the top images, showing the dynamic

flexibility of tube structures between 0 to 4 sec (0.4 sec per image). Scale bar, 5 µm. **d** Left, a relationship between contour length ($L$) and mean square of end-to-end distance ($<R^2>$) of the tube structures for estimation of the persistence length ($L_p$). The continued lines represent fitting curves (black for PuuE tube, red for actin filament) to experimental data (black open circle for PuuE tube, red cross mark for actin filament). Right, comparison of persistence length with cytoskeletal elements. PuuE tube (PT, black) and actin filaments (AF, red) were determined in this study (A wider range of plots is shown in Supplementary Fig. 7b). Intermediate filaments (IF, blue) and microtubules (MT, green) are taken from ref. 41,44, respectively. Source data are provided as a Source Data file.

conformations and may contribute to a deeper understanding of the intricate design principles of actin filaments.

As mentioned above, the helical conformation of tube structures is thought to arise from hydrophobic interactions, which are inherently sensitive to temperature and weaken at lower temperatures[49–51]. This led us to posit that alterations in temperature can serve as reversible switches for disassembly and reassembly. Notably, exposing the samples to 0 °C for 1 h suggested a dissociation of the helical conformations and hinted at a possible breakdown of the tubular structures (Fig. 4c and Supplementary Fig. 8d). Remarkably, when these disassembled samples were reintroduced to 30 °C for 24 h, the elongated tubular formations with helical conformations were restored. By grafting the D-loop, the tube structure could form helical conformations and acquired temperature-dependent reversibility. This thermal responsiveness parallels the behaviour of microtubules[21,22,52], underscoring the ability of this nature-inspired approach to mimic the dynamic properties of biomolecular assemblies in artificial protein design to create complex higher-order protein structures. This dual responsiveness (salt and temperature dependence) enhances the biomimetic potential of our design, which is a promising avenue for advanced applications in synthetic biology and materials science.

To determine the intricate helical configurations, structural analyses were performed using cryo-EM (Fig. 4d, Supplementary Fig. 9 and Supplementary Table 2). In addition to the inherent flexibility of the tube structure, the ability of tubes to form helical bundles introduces an additional layer of complexity to the structural analysis. This complexity is underscored by cryo-EM results, which render a detailed analysis of these higher-order structures particularly challenging. However, analysis of 2D class-averaged images of the helical structures revealed double and triple helical tubes, which was consistent with the nsTEM observation (Fig. 4b, d and Supplementary Figs. 8c, 9). Moreover, the tube structures forming these helices seem to show a thinner diameter of approximately 24 nm, which does not align with any of the original PuuE tubes with diameters starting at 28.6 nm (Fig. 3b and Supplementary Fig. 6). Therefore, an attempt was made to elucidate the characteristics of the tube structures forming the helical conformations by employing temperature-induced structural disassembly (Fig. 4c). The cryo-EM sample was initially prepared at 25 ± 1 °C to prevent disassembly; however, to unwind the helical structures, the sample was briefly chilled on ice for approximately 1 h. While chilling the sample on ice led to the unwinding of helical structures and partial disassembly of the tubes, the remaining short tubes were stable enough to allow cryo-EM analysis. By observing these chilled samples

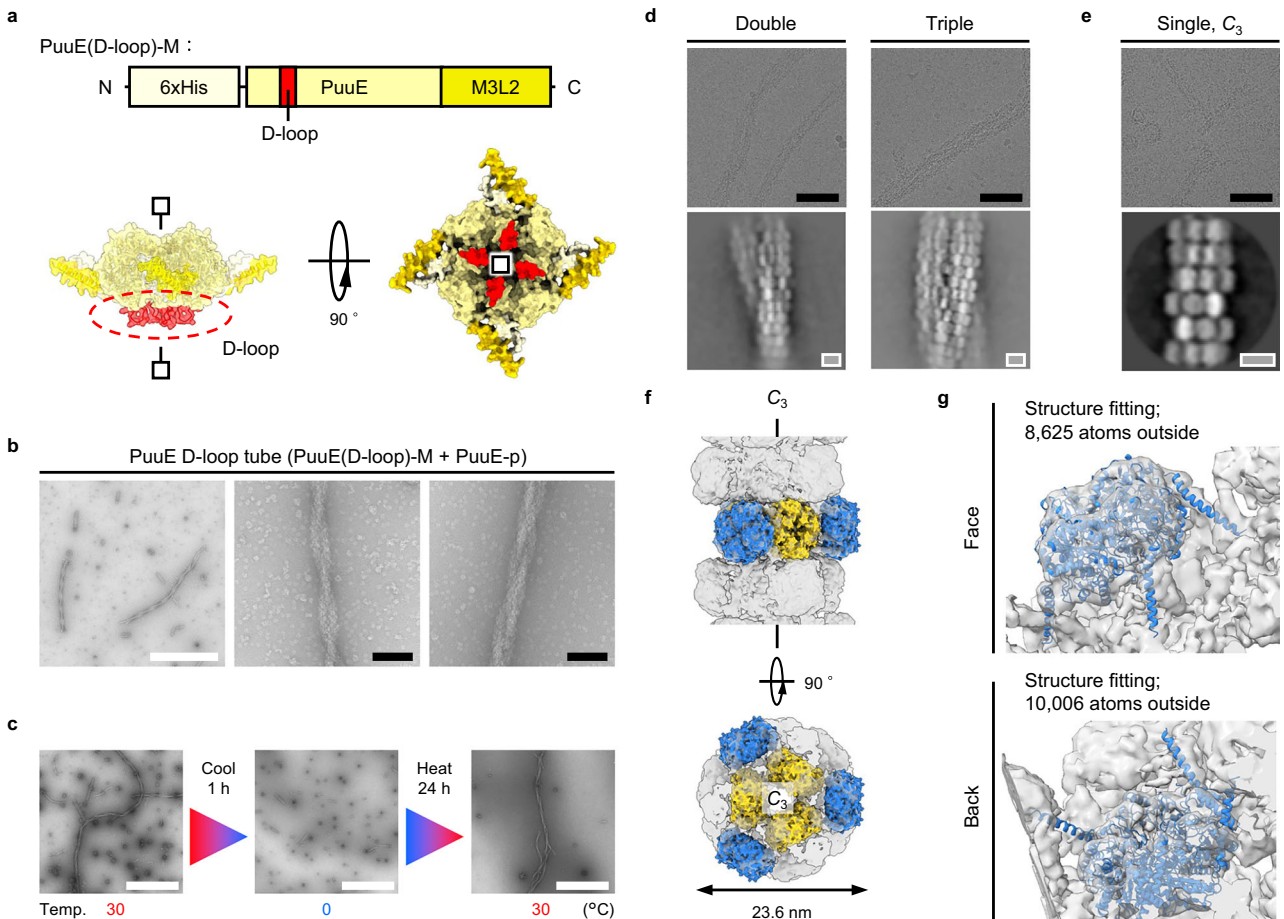

**Fig. 4 | Emulation of actin filament by D-loop grafting. a** Schematic representations of PuuE(D-loop)-M. The position of the D-loop graft (red) is indicated by protein sequence (top) and the AF2-predicted structure (bottom). Symmetry of protein is represented with lines and square markers. **b** nsTEM images of tubes with a helical conformation composed of PuuE(D-loop)-M and PuuE-p. The helical pattern of two (centre) or three (right) intertwined tubes is shown in the high-magnification image. **c** nsTEM images showing the reversibility of the tube structure with helical conformations by temperature change. Temp. indicates temperature. **d** Representative cryo-EM images (top) and 2D class-averaged images (bottom) of helical tube structures. **e** Representative cryo-EM image (top) and 2D class-averaged image of tube structure with $C_3$ symmetry. Tube structures with other symmetries found in this study are shown in Supplementary Fig. 10. **f** 3D reconstructed model of the tube structure with $C_3$ symmetry. For visibility, only the PuuE structure (PDB ID: 3CL6) is overlayed on the 3D reconstructed model. **g** Fitting of AF2-predicted model of PuuE-p into the 3D reconstructed model. The fitting results suggest that PuuE-p is unlikely to fit in the units located inside the tube structure; it is better accommodated by the units on the outside. The map density was normalised and visualised at a threshold of sigma ($\sigma$) value = 3.3 above average. Structure fitting was performed to count the atoms of the PuuE-p model outside the reconstructed 3D map of the $C_3$ tube structure at this threshold. Based on this prediction, the units in **f** are colour-coded as described in Fig. 1c. The Met-6xHis-TEVcs region of the PuuE-p model is not shown to improve visibility. Scale bars, 1 μm (white), 100 nm (black), 10 nm (grey).

with cryo-EM, we successfully identified a previously unobserved tube structure with $C_3$ symmetry (Fig. 4e, f and Supplementary Table 2), alongside the structures reported earlier (Supplementary Fig. 10). A comparison of this tube structure with $C_3$ symmetry with the structures forming the helical conformations indicated a match, suggesting that the tubes forming the helical conformation have indeed $C_3$ symmetry (Fig. 4d, e bottom). In addition, the diameter of approximately 23.6 nm, as determined by cryo-EM 3D reconstruction, corresponds to the tubes forming helical structures, further supporting these findings. Considering its inherent flexibility, it is challenging to reach a definitive conclusion, but further examination of the tube structure with $C_3$ symmetry suggests that PuuE-p is likely positioned on the outside (Fig. 4g and Supplementary Movie 4), consistent with the original PuuE tube structures (Fig. 3b and Supplementary Movie 2). This arrangement indicates that the D-loop of PuuE(D-loop)-M appears on the exterior of the tubes, which is crucial for forming helical structures not observed in PuuE tubes lacking the D-loop. The $C_3$ symmetry enhances the exposure of internal PuuE(D-loop)-M on the outer surface compared to structures with $C_4$ or higher symmetry, enabling hydrophobic interactions between tubes. Therefore, the formation of the $C_3$ symmetric tube structure likely facilitated the creation of the helical conformations. Furthermore, the lack of $C_3$ symmetry in PuuE tubes (Fig. 3a, b and Supplementary Fig. 6) suggests that they are unstable as single tubes without forming helical conformations. The formation of helical conformations may stabilise the structure with $C_3$ symmetry, as evidenced by its successful identification in tube structures with helical conformations. In addition, the temperature-induced degradation leading to the rapid collapse of tubes with $C_3$ symmetry suggests that helical structure stabilisation is essential for maintaining structural integrity under physiological conditions.

## Discussion

We introduce a pioneering approach that intricately combines the design and assembly of two distinct protein components, inspired by the complexity and versatility of natural protein assemblies. In contrast to previous studies that primarily relied on single-component systems[17,18], our method successfully demonstrates the formation of a higher-order tubular assembly composed of two distinguished protein

components. This dual-component design offers a significant advantage, enabling potential modulation of assembly dynamics and the integration of more complex functionalities. For example, one component acts as a carrier for the therapeutic drug and the other as a delivery system. Unlike single-component systems, this two-component approach allows precise control of the spatial arrangement and stoichiometric ratio of the components, which cannot be achieved with chemical modifications using a single component. In addition, it offers potential for environmental sensing systems, where assemblies respond to specific stimuli, such as pH or ionic strength, by undergoing structural reconfiguration. These systems could be engineered to detect biological or chemical signals in real time. Moreover, this adaptability could be leveraged to develop nanodevices for applications in biomedicine and bioengineering. Alternatively, different enzymes could be fused to each component and used as a scaffold for cascade reactions with an on-off system.

Notably, our study highlights the successful induction of helical conformations within these tube assemblies, resembling those observed in actin filaments. This was achieved by strategically incorporating the D-loop of actin into the assembly design, further showcasing the adaptability of our approach. While previous studies have engineered actin variants with modified D-loops[53,54], isolating the functional role of the D-loop in the context of native actin remains technically and biologically challenging due to the complex quaternary structure of actin and cooperative interactions between the D-loop and the four subunits. In contrast, our bottom-up approach enables the integration of this critical motif into a modular and structurally unrelated scaffold, allowing us to examine its contribution in a simplified and controllable environment. This strategy represents a conceptual shift in the use of artificial protein assemblies—not only to replicate biological structures, but also to functionally recontextualise natural motifs within redesigned protein frameworks. We believe this work serves as a proof-of-concept for leveraging artificial scaffolds to investigate and repurpose biologically essential structural elements. These findings pave the way for developing multi-functional protein-based materials with enhanced control over structural properties and potential applications in areas such as artificial cytoskeletons and dynamic transport systems.

This advance in protein assembly underscores the challenges of emulating the dynamic behaviours observed in biological systems. While our designed protein assemblies exhibit promising biomimicry in terms of flexibility and reversibility, their behaviour in biological environments remains to be validated. Factors such as macromolecular crowding, post-translational modifications, and cellular interactions significantly influence protein behaviour in vivo[55]. In the present study, we have succeeded in constructing tubular assemblies that are reminiscent of both actin filaments and microtubules. Interestingly, eukaryotic cells appear to have evolved to maintain cytoskeletal structures with specialised properties such as actin filaments and microtubules. Further modification of scaffolds and linkers of the tube components to create assemblies with diverse physical and dynamic properties and to study their behaviour in the cellular environment may reveal the general principles governing the functional specialisation of the eukaryotic cytoskeleton. Such studies could not only provide insights into the evolutionary and biological significance of cytoskeletal diversification but also contribute to the design of synthetic systems that better emulate cellular functions.

Our research extends the boundaries of protein assembly design and provides insights into its applications in synthetic biology and life sciences. By bridging the biological and materials sciences, demonstrating the transformative potential of integrating nature's complex systems into engineered solutions. Future investigations will likely build on this work, providing fundamental insights into protein assembly and opening pathways for innovations in regenerative medicine, adaptive biomaterials, and advanced nanotechnologies.

## Methods

### Plasmids and cloning
Primers for cloning and synthetic genes of N-terminal 6xHis-TEVcs-tagged PuuE-M and PuuE-p were purchased from Eurofins Genomics. PCRs were performed using the PrimeSTAR Max DNA Polymerase (Takara Bio) according to the manufacturer's protocol. Sizes of PCR products were verified using standard agarose gel electrophoresis. The In-Fusion Snap Assembly (Takara Bio) was used as the standard method for cloning according to the manufacturer's protocol, and each amplified gene fragment was ligated between the NdeI and BamHI multicloning sites of the pET11a expression vector (Novagen). Primers for cloning and a synthetic DNA fragment of the D-loop were purchased from Eurofins Genomics. The plasmid encoding N-terminal 6xHis-TEVcs-tagged PuuE(D-loop)-M was generated from the PuuE-M plasmid following the same procedures as above. All plasmids were amplified in E. coli strain DH5α (NIPPON GENE) and extracted using the NucleoSpin Plasmid EasyPure (MACHEREY-NAGEL) according to the manufacturer's protocol. DNA sequences were confirmed by a sequencing service (Eurofins Genomics). All primer sequences are shown in Supplementary Data 1.

### Protein expression and purification
The recombinant proteins were expressed using E. coli strain BL21 (DE3) (NIPPON GENE) co-transformed with a pGro7 chaperone plasmid (Takara Bio) and purified as follows. After transformation with plasmid DNA, colonies grown overnight on LB agar plates supplemented with 100 µg/mL ampicillin (Amp) and 20 µg/mL chloramphenicol (Crm) at 37 °C were picked to inoculate 5 mL of liquid LB-Amp-Crm broth and grown overnight at 37 °C and 200 rpm. Overnight cultures were diluted in 1 L of liquid LB-Amp-Crm broth supplemented with 0.5 mg/mL $_L$-arabinose and grown at 37 °C and 200 rpm until reaching an optical density at 600 nm of 0.6–0.8. Protein synthesis was induced by adding 0.1 mM isopropyl-β-$_D$-thiogalactopyranoside, and the cultures were grown at 16 °C for 16–20 h. Cells were harvested by centrifugation at 15,317 × $g$ and 4 °C for 5 min and then frozen at −80 °C. Cell pellets were thawed at 25 ± 1 °C, resuspended in 60 mL of ice-cold purification buffer (20 mM Tris-HCl, pH 8.0, containing 300 mM NaCl), and lysed using sonication (9 min with 1:2 on/off cycles and 70% amplitude; SFX250, Branson) on ice. Cell debris was cleared by centrifugation at 15,317 × $g$ and 4 °C for 30 min. The supernatant (i.e., crude protein) was filtered through a 0.45 µm pore size membrane filter (Merck), applied onto HisTrap FF crude column (Cytiva) pre-equilibrated with the purification buffer and washed with 5 column volumes of 2% elution buffer (20 mM Tris-HCl, pH 8.0, containing 300 mM NaCl and 1 M imidazole; 2% means 20 mM imidazole). 6xHis-TEVcs-tagged proteins were eluted with 10 column volumes of elution buffer with a linear gradient of 2–40% (i.e., 20–400 mM imidazole). The fractions containing the proteins confirmed by means of UV absorption and SDS-PAGE were again collected and dialysed against a 50-fold volume of NaCl (+) or NaCl (−) buffer (50 mM Tris-HCl, pH 8.0, containing ±100 mM NaCl and 0.5 mM EDTA) at 4 °C twice. Each of the purified proteins was concentrated by an Amicon Ultra centrifugal filter unit (Merck) with an appropriate molecular weight cutoff, followed by filtration through a 0.45 µm pore size membrane filter (Merck). Protein concentration was determined by absorbance measurements at 280 nm using a NanoDrop OneC spectrophotometer (Thermo Scientific). The molar extinction coefficients at 280 nm for the proteins were calculated from the basis of amino acid composition[56]. The concentrated proteins were frozen in liquid nitrogen and stored at −80 °C before experiments.

### Sample preparation
All proteins were thawed immediately before tube formation experiments on ice. Each sample was prepared in a 1.5 mL microtube using an appropriate buffer to adjust the concentration described in the

manuscript and the volume to 200 μL at 25 ± 1 °C. Except for the NaCl concentration-dependent experiments, NaCl (+) protein stock solution and buffer were used. For the NaCl concentration-dependent experiments, 50 mM Tris-HCl (pH 8.0), 1 M NaCl, and 0.5 mM EDTA were used in addition to NaCl (−) protein stock solution and buffer. Incubation of the samples was carried out using a ThermoMixer C (Eppendorf) or a MATRIX Orbital Delta Plus (IKA) with shaking of 300 rpm at the temperature described in the manuscript. For the disassembly and reassembly experiments, buffer substitution procedures were conducted using NaCl (−) and NaCl (+) buffer, respectively, with Microcon 50 centrifugal filter units (Merck) according to the manufacturer's protocol four times at each step.

### Negative-stain transmission electron microscopy (nsTEM)

A naked G600TT copper grid (Nisshin EM) was carbon-coated using a VE-2030 (VACUUM DEVICE). The grid was glow-discharged using a PIB-10 (VACUUM DEVICE). Then, a 5 μL aliquot of the sample solution was placed on the grid for 1 min, and the remaining solution was removed with filter paper (No. 2, ADVANTEC), followed by rinsing thrice with a 5 μL aliquot of Milli-Q water. After blotting off the water with filter paper, the sample was stained briefly with a 3 μL aliquot of 2% (w/v) uranyl acetate solution three times. The remaining solution was removed with filter paper, and the grid was dried on the bench-top. TEM observation was performed using a transmission electron microscope HT-7700 (Hitachi High-Tech) with an acceleration voltage of 80 kV. The images were recorded using HT-7700 control software version 02.22 (Hitachi High-Tech).

### Tube length analysis

Hundreds of discriminable tubes were picked up manually on 5k-magnification nsTEM images. The tube lengths were calculated as half of the perimeter analysed with ImageJ (Fiji) version 1.53[57]. The plots were drawn by selecting 150 tubes from the longer lengths using Igor Pro 9 version 9.0.5.1 (WaveMetrics).

### Circular dichroism (CD) spectrum measurements

All proteins were thawed immediately before CD measurements on ice. Each sample was prepared in a 1.5 mL microtube using NaCl (+) buffer to adjust the concentration to 2.5 μM and the volume to 200 μL at 25 ± 1 °C. Far-UV CD spectra were obtained at a wavelength of 200−250 nm using a J-1100 spectropolarimeter (JASCO) with a quartz cell with a light path of 1 mm. Thermal denaturation was performed at a temperature change rate of 1 °C/min. The CD spectral data were collected using Spectra Manager version 2.5 (JASCO). All CD data were expressed as mean residue ellipticity. The $T_m$ of each protein was calculated from the thermal denaturation curve at a wavelength of 222 nm by sigmoid fitting using Igor Pro 9.

### TEV protease treatments

TEV protease (FUJIFILM Wako Pure Chemical) was used to cleave the His-tags. All proteins were thawed immediately before treatment on ice. Cleavage reactions were carried out according to the manufacturer's protocol and verified by SDS-PAGE. After digestion, residual intact proteins and TEV protease were removed using a HisTrap HP column (Cytiva). The cleaved proteins were then buffer-exchanged into either NaCl (+) or NaCl (−) buffer using Amicon Ultra centrifugal filter units with appropriate molecular weight cutoffs. Protein solutions were subsequently filtered and their concentrations determined as described in the "Protein expression and purification" section. The final cleaved proteins were frozen in liquid nitrogen and stored at − 80 °C until use.

### Cryo-EM sample preparation and data collection

All proteins were thawed immediately before tube formation on ice. Each sample was prepared in a 1.5 mL microtube using NaCl (+) buffer

to adjust the concentration to 12.5 μM and the volume to 200 μL at 25 ± 1 °C. Incubation of the samples was carried out as described above for 24 h at 40 °C for the PuuE tube and 30 °C for the PuuE D-loop tube, and then the samples were provided for grid preparation. For unwinding the helical structures of the PuuE D-loop tube, additional incubation was carried out for 1 h on ice immediately before grid preparation.

The PuuE D-loop tube sample prepared at 25 ± 1 °C was used at the original concentration. In contrast, the PuuE tube and the PuuE D-loop tube preincubated on ice were diluted to one-third and one-sixth of their original concentrations, respectively. Quantifoil R1.2/1.3 Cu 300 grids coated with a holey carbon film (Quantifoil) were treated for hydrophilisation using a JEC-3000FC Auto Fine Coater (JEOL) at 20 Pa and 10 mA for 30 s. Subsequently, 2.5 μL aliquots of the respective diluted samples were applied to the prepared grids. After blotting off excess solution, the grids were rapidly immersed in liquid ethane for vitrification using a Vitrobot Mark IV (Thermo Fisher Scientific). Vitrobot was set at 4 °C and 100% humidity for PuuE tube and PuuE D-loop tube samples preincubated on ice, and 25 °C and 100% humidity for PuuE D-loop tube samples prepared at 25 ± 1 °C.

Sample screening and data acquisition were performed using a Glacios cryo-transmission electron microscope (Thermo Fisher Scientific) operated at an accelerated voltage of 200 kV, equipped with a Falcon4EC camera, at the Institute of Life and Medical Sciences, Kyoto University. Images were automatically acquired using the EPU software version 2.14 and 3.3 as movies with nominal magnifications and corresponding calibrated pixel sizes of 120,000 × (1.22 Å/pixel) for the PuuE tube sample, and 150,000 × (0.925 Å/pixel) for PuuE D-loop tube samples.

### Cryo-EM image processing

Image analysis was conducted using similar workflows for each dataset of the three samples with the software packages RELION 5.0beta[58,59] and cryoSPARC version 4.1.1[60].

For the PuuE tube sample, 4346 movies were subjected to motion correction using RELION's algorithm, and the contrast transfer function (CTF) was estimated using CTFFIND4 version 4.1.14[61]. Tube coordinates were manually registered, and 709,722 segments were extracted with 3 × binning into 260 × 260-pixel boxes (approximately 950 × 950 Å) with an inter-box spacing of 80 Å. The extracted segments were subjected to two rounds of 2D classification, and the resulting class averages were visually inspected to categorise the segments based on tube diameters. In parallel, an additional round of 2D classification with 10 classes was performed to assess the structural diversity of the tubes roughly. Each subset of segments, categorised by diameter, was then re-extracted and subjected to further 2D classifications to remove junk images. Initial 3D reconstructions were carried out using the 'Helical Refinement' algorithm in cryoSPARC. A featureless hollow cylinder was used as the starting reference for the first iteration in refinement to avoid reference bias, with inner and outer diameters approximately measured from 2D class averages. The resulting helical structure from this initial refinement job was then used as the reference volumes for subsequent analyses in RELION. After importing these volumes into RELION, 3D classification with symmetry search was performed on each subset without imposing symmetry ($C_1$) using the initial 3D reconstruction as a reference. Finally, 3D refinement was carried out for the subsets with the three smallest diameters, applying $C_4$, $C_5$, and $C_6$ symmetries, respectively. The subsets with larger diameters exhibited significant heterogeneity and did not yield reliable 3D reconstructions.

For the PuuE D-loop tube sample prepared at 25 ± 1 °C, 4871 movies were motion-corrected and CTF-estimated using RELION and CTFFIND4, respectively. A total of 126,987 segments were extracted with 5× binning into 320 × 320-pixel or 640 × 640-pixel segmented boxes (1480 × 1480 or 2960 × 2960 Å) with an inter-box spacing of

60 Å. The extracted segments were subjected to six rounds of 2D classification, yielding class averages displaying single tube architectures, and double and triple helical tube architectures.

For the PuuE D-loop tube sample, preincubated on ice to unwind the helical structures, 4346 movies were subjected to motion correction and CTF estimation. A total of 709,722 segments were extracted with $3\times$ binning into $360 \times 360$-pixel boxes (approximately $1000 \times 1000$ Å) with an inter-box spacing of 80 Å. The extracted segments were subjected to two rounds of 2D classification, and the resulting class averages were visually inspected to categorise the segments based on tube diameters. In parallel, two rounds of 2D classification were performed to assess the structural diversity of the tubes. Each subset of segments, categorised by diameter, was re-extracted and subjected to further 2D classifications to remove junk images. Initial 3D reconstructions and 3D classification were performed similarly with the PuuE tube. During the 3D classification of the initially selected $C_5$ tube subset, $C_6$ tubes were found to be present and were subsequently combined with the $C_6$ tube subset from the 2D classification. Finally, 3D refinement was carried out for the subsets with the four smallest diameters, applying $C_3$, $C_4$, $C_5$, and $C_6$ symmetries, respectively. As observed in the PuuE dataset, the subsets with larger diameters displayed considerable heterogeneity and failed to yield reliable 3D reconstructions.

For both samples, 2D reprojections were generated using the *relion_project* command from the refined 3D reconstructions and compared with the 2D class averages to assess potential symmetry-related artifacts. For 3D classification and refinement, the initial volumes were low-pass filtered to 40 Å resolution to avoid reference-based artifacts. Particle binning, applied during initial extraction, was maintained throughout subsequent classifications and 3D reconstructions. This approach was adopted as the estimated resolution of the final reconstructions did not reach the Nyquist limit of the binned images. To validate this method, we also performed reconstructions using boxes with the original pixel size, particularly for the D-loop $C_3$ tube. These efforts yielded similar results to those presented, with comparable helical parameters and resolutions. Our comprehensive exploration confirms that the presented reconstructions represent the best achievable resolutions for these structures, given the current data.

Detailed image processing workflows are depicted in Supplementary Figs. 6, 9, and 10.

### Fluorescent labelling

Tube formation was conducted as described above under the optimised condition described in the manuscript. Labelling reaction was achieved by adding Alexa Fluor 488 succinimidyl ester (Thermo Fisher Scientific) dissolved in dimethyl sulfoxide to the tube solution at a final concentration of 0.7 mM. The reaction was then incubated at $25 \pm 1\,°C$ for 1 h with gentle shaking under shading. The excess dye was removed using NaCl (+) buffer with Microcon 300 centrifugal filter units (Merck) according to the manufacturer's protocol four times. The labelled tubes were then stored under shading at $25 \pm 1\,°C$ until further experiments.

### Fluorescence microscopy

An observation chamber was assembled by placing two double-sided tapes (thickness $\sim100\,\mu m$) onto a silicone-coated coverslip ($24 \times 36\,mm^2$, thickness No. 1; Matsunami) with another coverslip ($18 \times 18\,mm^2$, thickness No. 1; Matsunami) on top. To passivate the surface of the coverslips against nonspecific adhesion of protein, the chamber was filled with 10 mg mL$^{-1}$ of Pluronic F-127 (Sigma-Aldrich) dissolved in distilled water for more than 10 min at 25 °C. After washing out Pluronic F-127 solution with 5 chamber volumes of NaCl (+) buffer, the chamber was filled with TIRFM buffer (50 mM Tris-HCl, pH 8.0, 100 mM NaCl, 0.5 mM EDTA, 0.2% (w/v) methylcellulose (1500 cP,

Wako), 1 mM DTT, 2 mM Trolox). Next, the Alexa488-tube solution was diluted to 1/10 in NaCl (+) solution, and further diluted to 1/10 (final 1/100 dilution) in TIRFM buffer. Then, the diluted tube solution was perfused into the observation chamber and sealed by Valap to prevent flow. The fluorescence images of tube structures were acquired at intervals of 40 ms with an inverted microscope (IX-71, Olympus) equipped with a 60 × objective lens (PlanApo NA 1.45 oil, Olympus), an EMCCD camera (iXon3, Andor Technology), and an excitation laser with the wavelength at 488 nm (OBIS 488-60-LS, COHERENT). All observations were performed at $25 \pm 1\,°C$.

### Mechanical property analysis

The persistence length of the tube structures was estimated as follows. First, the fluorescence images were converted to 8-bit images using the ImageJ (Fiji) version 1.54 function. Then, the skeletons of the tube structures were tracked using the ImageJ plugin, JFilament version 1.02[62]. Distances between adjacent nodes composing the skeletons were set as 1 pixel. Next, the contour length ($L$) and end-to-end distance ($R$) of the tube structures at each frame were calculated using the coordinates of the nodes with custom-written Python scripts (Python version 3.10.12). The mean square of $R$ ($\langle R^2 \rangle$) of each tube structure was calculated by averaging $R^2$ along 100–200 frames.

$\langle R^2 \rangle$ and $L$ follow the following Eq. (1) when the shape fluctuation is driven thermally[39]:

$$\left\langle R^2 \right\rangle = 4L_p^2 \left[ 2\exp\left(-L/2L_p\right) - 2 + L/L_p \right] \tag{1}$$

where $L_p$ is the persistence length of the tube structure. The $L_p$ values of the tube structures were estimated by fitting this equation to the experimental data using the 'curve_fit' function of the Python package 'scipy. optimize' (SciPy version 1.11.4). $L_p$ of actin filaments was estimated by the same analysis. Totally, 55 tube structures and 37 actin filaments were analysed.

### Molecular modelling

All predicted protein structures were generated by AlphaFold 2.2 or 2.3 multimer-mode (DeepMind)[14,15]. Predicted local distance difference test plots of PuuE-M and PuuE-p 'AF2 ranked_0 models' were generated from respective .pkl files using af2visualizer.py script[63]. Cartoon models of the proteins were drawn using PyMOL version 2.5 (Schrödinger)[64] and UCSF ChimeraX version 1.7 (UCSF RBVI and NIH)[65]. Isoelectric points of PuuE-M and PuuE-p were calculated from the basis of amino acid composition[66]. Surface electrostatic potential of PuuE-M and PuuE-p were calculated and drawn by APBS (version 3.4.1) and PDB2PQR (version 3.5.2) plug-in[67] on PyMOL. Surface hydrophobicity of PuuE was drawn using the Color_h script (PyMOL Wiki) based on the hydrophobicity scale[68].

### Statistics and reproducibility

For the analysis of PuuE tube disassembly and reassembly, a two-sided Welch's $t$ test was performed to test the null hypothesis, and exact $P$ values are provided in the figure legend. The PuuE tube formation experiment under optimal conditions (100 mM NaCl, 40 °C, 24 h) was repeated using five biological replicates, with similar results confirmed by nsTEM. Each of the PuuE tube formation experiments under non-optimal conditions was performed once as part of preliminary screening efforts to explore and optimise conditions for efficient tube assembly. The PuuE tube disassembly and reassembly experiment was performed using two biological replicates, yielding consistent results confirmed by nsTEM. The PuuE D-loop tube formation experiment and the cold-induced disassembly experiment were each conducted with four biological replicates, showing consistent outcomes confirmed by nsTEM. The reassembly of the PuuE D-loop tube was performed using two technical replicates, with similar results confirmed by nsTEM.

**Reporting summary**

Further information on research design is available in the Nature Portfolio Reporting Summary linked to this article.

## Data availability

Source data are provided in this paper. The cryo-EM structures have been deposited in the Electron Microscopy Data Bank (EMDB) under the following accession codes: EMD-60617, EMD-60618, EMD-60619 for PuuE tubes with $C_4$, $C_5$, and $C_6$ symmetry, respectively; and EMD-60620, EMD-60621, EMD-60622, EMD-60623 for the D-loop tubes with $C_3$, $C_4$, $C_5$, and $C_6$ symmetry, respectively. The source data supporting the findings of this study are available on figshare and will be publicly accessible upon publication via the following https://doi.org/10.6084/m9.figshare.28397147. All additional data supporting the findings of this study are available within the article, in the Supplementary Information, or from the corresponding author upon request.

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

## Acknowledgements

This work was supported by JSPS KAKENHI (grant nos. 19H02832, 19K22253, and 21H05116 to Yuta Suzuki (Y. Suzuki); 21H05117 to Y. Suzuki and Yukihiko Sugita (Y. Sugita); and 20K22628, 21J00530, and 22KJ1644 to M.N.), JST PRESTO (grant no. JPMJPR22A7 to Y. Suzuki and JPMJPR20ED to M.M.), Takeda Science Foundation to Y. Suzuki, Chubei Itoh Foundation to Y. Suzuki, and The Hakubi Center for Advanced Research to Y. Sugita, M.M., and Y. Suzuki.

## Author contributions

Y. Suzuki directed the project. Y. Suzuki and M.N. conceived and designed the overall study. M.N. conducted experimental work with contributions from Y. Suzuki, Y. Sugita, and Y.Y. Y. Sugita and M.N. performed cryo-EM data collection and analysed data. Y.Y. conducted TIRFM experiments, and Y.Y. and M.M. analysed mechanical properties. Y. Suzuki and M.N. wrote the manuscript with contributions from Y. Sugita, Y.Y., and M.M.

## Competing interests

Y. Suzuki and M.N. are inventors on a Japanese patent application related to the protein assembly structures described in this study (Japanese Patent Application No. 2024-65760, filed on April 15, 2024). The remaining authors declare no competing interests.
