## [Transparent Peer Review file · Nature Communications]

Protein design of two-component tubular assemblies similar to cytoskeletons

Corresponding Author: Dr Yuta Suzuki

Version 0:

Reviewer comments:

Reviewer #1

(Remarks to the Author)

[Editorial note: Please find Reviewer #1's comments at the end of this document.]

Reviewer #2

(Remarks to the Author)

Reviewer #3

(Remarks to the Author)

The manuscript by Noji et al describes the design and assembly of two-component protein modules into long tubes. Those tubes have high persistent length and some variability in the diameter and self-assemble in a defined temperature and ionic strength range. They are formed by the fusion of two natural domains, one of them common and the second from two interacting domains. Interestingly, the transplantation of the actin D loop results in the formation of a helical assembly composed of two of three filaments with narrower diameters, that are not stable by themselves but require helical suprastructural assembly. The role of just D loop is quite unexpected and may shed some light into its role in the actin assembly. CryoEM analysis provides good models for the structure of filaments.

However, this is not the first designed tubular assembly, as claimed. The authors should include a reference to Shen et al., Science 2018, from the Baker group, which reported on the design of protein filaments several years ago (DOI: 10.1126/science.aau3775).

Apart from that the paper is well-presented using several techniques to analyze the assembly and properties of designed tubes.

The authors suggest that the P_uE is positioned on the outside. This could be in principle tested by antibodies against a peptide tag that should be able to access the exposed domains but not protected ones.

Additionally, it would be interesting to know the persistence length of the helical assemblies, particularly whether they are longer than the filaments without the D loop.

Also, could they provide information on the interactions between the D loops in the intertwined tubes, is the conformation (at least according to the model) similar to the conformation in actin filaments ?

Reviewer #4

(Remarks to the Author)

This manuscript describes the design, assembly and characterization of repeat tubular structures. A flexible dimer component was grafted on a scaffold protein and the two component system, when mixed, forms fibers in vitro. The authors describe their design principles and outline changes to the fibers that seem to depend on salt concentration and temperature. Additionally, they introduce another graft resulting in the formation of higher order helical fibers.

This work is interesting and the manuscript is fairly well-written and clear. I have some concerns and points for the authors to address and clarify:

Major points:

- the acronym used (NIPAD): it is not clear why the methods employed in this study requires this naming scheme as there was no development of new engineering methods or software.
- Cryo processing: please add more information, including images of initial models used and how C1 symmetric reconstructions deviated from the final symmetric reconstructions (if any). This is crucial to understand potential symmetric artifacts at low resolutions, especially for the higher order assemblies.
- methods were not clear as to when binning was reversed during cryoem processing. Was binning reversed for the final resolution calculations?
- it is not clear why this design results in tubes and not 2D or 3D crystals or closed c-symmetric oligomers or even large vesicles. Were any observed?
- Looking at the final maps, it seems that additional interactions occur between the scaffold proteins than was originally intended: please discuss
- Figures need greater clarity - for example, figure 1 should more clearly show the interactions that form fibers: it was not easy to understand what the authors see in the design that would create a fiber over other assemblies. Additionally, the symmetric axes should be well defined and repeat units outlined
- Cryo processing: did the authors try symmetric expansion and other techniques such as focused refinement to try and discern the interfaces or get higher resolution refinements that better account for dynamics?
- AlphaFold: have the authors tried to understand the assembly using modeling of multiple subunits?
- Discussion: more detail on the potential applications of these fibers would be beneficial
- The authors claim that the design of fibers "remains elusive": fibers have been designed before, such as Bethel et al. 2023 (nature chemistry). Previous fiber studies should be discussed.
- Control over assembly: apart from the environmental factors discussed, do the authors have finer control over the fiber assembly and oligomeric states by design?
- D-loop grafted cryoem: the sample was left on ice for 1hr before freezing and fibers were observed, however, this is in contrast to the negative-stain experiments in Fig4c and ExtFig7d, where there are no observable fibers at 0c. Please clarify

Minor points:

- "cryo-electron": the hyphen should not be between cryo and electron.
- Figure 4g: number of atoms outside contour is not very useful without knowing the threshold used and where the greatest deviation between map and model lies
- the role of M3L2/p66a in the MBD2-NuRD complex is not described

Version 1:

Reviewer comments:

Reviewer #1

(Remarks to the Author)

The reviewer's comments and concerns have been adequately addressed. Although the additional experiments involving cryo-EM and SAXS to investigate the formation mechanisms were not performed, the dynamic and responsive properties demonstrated in this study represent a significant advancement in the field of protein assembly. This is supported by recent reports highlighting the importance of such properties (e.g., Nat. Nanotechnol. 2024, 19, 1016; Nat. Chem. Biol., 2025, doi.org/10.1038/s41589-024-01811-1; Nat. Synth., 2025, doi.org/10.1038/s44160-024-00726-y; Chem, 2025, doi.org/10.1016/j.chempr.2024.102407). In its current form, this manuscript meets the standards for publication in Nature Communications.

Reviewer #2

(Remarks to the Author)

Reviewer #3

(Remarks to the Author)

Authors have mainly modified the text of the manuscript without additional data or insights. Although there are still some

unanswered questions (e.g. regarding the role of the “D-loop”) and challenges, the manuscript nevertheless describes an interesting design of protein fibers composed of two components. The authors built their assembly based on natural homotetramers and modified them by weak heterodimeric domains to make a two-component assembly. This approach could be applied in principle to many other building blocks and therefore represents a new design platform.

While authors did not report success by modeling the assembly using AF2, I Reviewer #3) obtained formation of 4+4 assembly linked through the CC peptides using AF3 algorithm (image of the models added). 4+4 assembly represents a dead end as it is not able to form larger filaments but may represent an intermediate in equilibrium with filamentous assemblies. Authors may invest a bit more effort into modeling and explanation of the putative mechanisms.

One issue that might be discussed is whether the two components need to be produced separately before mixing, to avoid formation of heterotetramers, which might hinder the formation of filaments. Did they ever try the production of both components at the same time?

As the authors concur that they did not demonstrate any cytoskeletal function and removed the phrase “reminiscent of the cytoskeleton”, they might as well remove the reference for the cytoskeleton from the title.

[Editorial note: images of the models are not included in this file due to copyright concerns.]

Reviewer #4

(Remarks to the Author)

Thank you for your efforts in addressing my concerns. Most concerns have been addressed, however, I have follow up comments below:

Comment 2:

You have not included images of the initial models used, only a description. Including the initial models will help compare to the final reconstruction.

Comment 5:

I think your text is reasonable to include, however, you should actually check how much your tag is influencing the formation or stability of the tubes by cutting the protein using tev. The construct already has this feature.

Comment 7:

I don't think I fully agree. While you may not reach high resolution and for some of your reconstructions I agree, you may not get more information, for others you could get more resolution leading to enough resolution to discern helices (such as your 9.7Å reconstruction of the c3 tube). Your interface would be revealed to greater detail and would help validate your design.

Minor point 2:

Again, this is not effective without knowing the threshold used. Different values will "contain" different amounts of atoms. The new term of "visual inspection" is not good to use either way.

Version 2:

Reviewer comments:

Reviewer #3

(Remarks to the Author)

I find the manuscript appropriate for publication due to its novelty and potential interest. I still believe that the assembly they produced is not related to the cytoskeleton. Still, I don't insist on removing the word from the title, where its function seems to be mainly to attract the attention of the readers.

Reviewer #4

(Remarks to the Author)

Thank you for including extra information. I do have the following comments for the Authors, which are mainly comments from a previous review that the Author's have not fully addressed yet.

I think clearing the following can be done and given that, I think this manuscript would be ready for publication.

My comments:

1. My previous comment of "You have not included images of the initial models used, only a description. Including the initial models will help compare to the final reconstruction".

Thank you for adding these, but they contradict your methods section where you state that the initial models were hollow cylinders as these initial models are not generic hollow cylinders. Can you please clarify and update the text and/or figures appropriately?

You don't mention if the models were low pass filtered for refinement. Please address.

2. My previous comment "I think your text is reasonable to include, however, you should actually check how much your tag is influencing the formation or stability of the tubes by cutting the protein using tev. The construct already has this feature"

I can understand your reasoning for keeping the tag, however, do you see any formation of the fibers after tev cleavage? Even with low concentrations, you can easily validate this using EM images and rule out of the effect of the tag.

3. My previous comment of "I don't think I fully agree. While you may not reach high resolution and for some of your reconstructions I agree, you may not get more information, for others you could get more resolution leading to enough resolution to discern helices (such as your 9.7Å reconstruction of the c3 tube). Your interface would be revealed to greater detail and would help validate your design."

The author's did not address this point as they mixed up my suggestion with another suggestion regarding binning and using the full pixel size. My comment was regarding using symmetry expansion or masking/local refinements to potentially improve the resolution of some of your reconstructions, mainly the one at 9.7Å. This can be done using the binned particles. I think this could improve the resolution potentially to the 6-8 range where helices would be better defined and better validate your designed interface.

Version 3:

Reviewer comments:

Reviewer #4

(Remarks to the Author)

Thank you for looking fully into my questions. I only have additional comments in regard to "Comment 2", tev cleavage of his-tag.

The aim of this experiment was to check that the tubular assembly is not influenced by the his-tag and based on your results, it seems to be. Based on this, I suggest the authors discuss this in the manuscript. Are the assemblies consistent in size and morphology to pre-cleaved assemblies?

Dear Dr. Editor,

First of all, my coworkers and I would like to thank you for handling our paper and the four reviewers for their time and thoughtful comments. We were very pleased to see that the reviewers described our work as "interesting and fairly well-written", "providing good models for the structure of filaments", and "achieving an exciting system of two-component protein assembly". They also acknowledged that the study offers "unique insights into protein assembly under biomimetic conditions" and highlighted its "potential significance for understanding actin assembly mechanisms." These positive evaluations have greatly encouraged us and affirmed the value of our research. Yet, they also raised several issues and had comments that require to be addressed for further consideration of our manuscript. Accordingly, we now submit a substantially revised version of our manuscript that addresses these issues and improvements in the main text and supporting materials with additional experiments. We also submit an additional version of the Main Text wherein the changes are highlighted in yellow for clarity. In addition to these corrections, we have also adjusted the manuscript format to meet the guidelines of Nature Communications.

RESPONSE TO REVIEWER #1

We greatly appreciate the reviewer's time and thoughtful comments on our manuscript. The feedback provided has helped us refine our arguments and clarify the unique aspects of our study. Below, we address each point raised by the reviewer in detail.

Comment 1: The main concept of Nature-Inspired Protein Assembly Design is ambiguous and unconvincing, as many works have established reversible protein nanotubes that respond to biomimetic conditions (Nat. Commun. 2022, 13, 5424.; J. Am. Chem. Soc. 2018, 140, 1, 26.; Nat. Chem. 2013, 5, 613.; J. Am. Chem. Soc. 2013, 135, 31, 11509.; ACS Catal. 2020, 10, 9735.). Also, in this work, a meaningful application of these nanotubes have yet to be demonstrated and explored.

Response: We thank the reviewer for highlighting this concern. The works cited in comment 1 indeed represent significant advancements in the chemical or hybrid control of protein assemblies through modifications with organic compounds, nucleic acids, or polymers. However, our study focuses on achieving reversible assembly and disassembly of protein nanotubes using only proteins, directly inspired by biological processes. This distinguishes our work from the cited studies, as our approach does not rely on external chemical modifications but rather utilizes inherent protein properties to achieve biomimetic assembly.

Given the ongoing challenges in controlling the assembly and disassembly of artificial protein assemblies, we believe our work offers both novelty and uniqueness in addressing this critical area.

Comment 2: The definition of two-component tube structures is controversial. Here, the main scaffold of P_{uu}E-M and P_{uu}E-p is the same, but the two-component protein assembly should be two independent ingredients (Nature 2014,510, 103.; Science 2016, 353, 389.; Nature 2021, 589, 295 468-473).

Response: We understand the reviewer's concern regarding the definition of two-component assemblies. In our work, the two components, P_{uu}E-M and P_{uu}E-p, individually do not form tubular structures. However, when mixed, they synergistically self-assemble into well-defined tubular structures. This cooperative behavior clearly demonstrates that both components are essential for the formation of the final structure, meeting the criteria of "a two-component system" as commonly defined in structural biology.

Unlike single-component systems reported in prior studies, such as Reviewer 3 and 4 indicated (Shen et al., Science, 2018 and Bethel et al., Nat. Chem., 2023), where tubes are formed by a single protein component, our system highlights the necessity of two distinct components that cannot independently achieve the same structural outcome. This distinction is a critical aspect of our study and contributes to its novelty.

Additionally, related studies involving two-component cage (Nature 510, 103-108 (2014) and Science 353, 389-394 (2016)) or sheet structures (Nature 589, 468-473 (2021)) as mentioned in comment 2 often rely on similar principles of cooperative interaction, where individual components alone do not form the intended structure. To our knowledge, our work represents the first demonstration of a two-component system forming tubular structures under these criteria, underscoring its originality and significance in the field. Moreover, our design enables reversible structural control of the assembled tubes, a feature not achieved in the designs presented in previous studies as mentioned above. This represents an additional advantage of our approach, further emphasizing the significance of our work. To clarify this point, we have included additional sentences in the main text (Lines 77–80 and Lines 262-266) and added references 18 and 19 to support the discussion.

[editorial note: unpublished data has been redacted]

Comment 3 & 4: Although the structures of protein nanotubes are supported by 3D reconstruction, the underlying mechanisms of the formation of these structures have not yet been explored and explained. Why do different nanotubes emerge under the same assembly conditions? Could certain conditions generate dominant species? Tracking the formation of nanotubes under thermally stable conditions is important to gain insight into the self-assembly mechanism using cryo-EM, SAXS, or other characterization methods.

Responses: We appreciate the reviewer's insightful comments regarding the formation mechanisms and characterization of protein nanotubes. As both comments 3 and 4 pertain to the structural formation of nanotubes, we have provided a single comprehensive response below.

We hypothesize that the formation of nanotubes with different diameters can be attributed to the flexibility of the connection sites on the protein surface. Similar findings have been reported in the literature, such as in studies on flexible protein designs that lead to varied structural outcomes (e.g., *Nat. Chem.* 6, 1065–1071 (2014) and *J. Am. Chem. Soc.* 137, 11285–11293 (2015)). In our study, the inherent flexibility of our design likely resulted in structural diversity. However, due to the resolution limitations of cryo-EM, knowledge of detailed structures is speculative. Unfortunately, this limitation is further complicated by the flexibility of the assembled structure, making it inherently difficult to achieve higher resolution for such specific assemblies (Lines 162-164 and Supplementary Movie 1). As a response to these comments, we have added this clarification in the revised manuscript under the section “Diversity and Flexibility of Tubes” (Lines 173-176) and added references 38 and 39.

In this study, the key factor for forming assemblies is the interaction between M3L2 and p66 α . Upon revisiting the data to examine how temperature and salt concentration influence the distribution of tube diameters, the results suggested a tendency for the diameter distribution to shift toward narrower tubes at around 25 °C compared to 35 °C and 40 °C. To further investigate this indication, we conducted additional experiments to observe the diameter of tube structures at 25 °C. However, no significant changes were observed as we expected. Given the flexible design principles of this system, achieving convergence to a single, uniform diameter by altering conditions appears challenging. For future studies, we plan to explore the possibility of controlling tube diameters using nucleic acids and etc. as scaffolds. However, incorporating different polymers would deviate from the scope of this research as discussed on comment 1, and we therefore intend to report such developments as follow-up studies.

While SAXS is a powerful technique for analyzing self-assembly mechanisms, we consider it less suitable for our system due to the coexistence of tubes with various diameters and lengths, which would complicate data interpretation. Moreover, based on the resolution and details achieved with cryo-EM, we determined that SAXS would not provide additional insights beyond what has already been obtained.

It is worth noting that, as highlighted in the cited studies, the creation of artificial protein assemblies has made significant progress, but detailed mechanistic understanding of their formation processes remains elusive. We recognize this as a critical challenge for the field and an important avenue for future exploration.

Comment 5: The overall electrostatic potential of PuuE-M and PuuE-p should be shown in Extended Data Fig. 5, especially since the salt concentration range for salting-in and salting-out is very narrow compared to most proteins.

Response: We appreciate the reviewer's suggestion regarding the inclusion of electrostatic potential data. As suggested, we have added the electrostatic potential of PuuE-M and PuuE-p to Supplementary Fig. 5a to provide a more comprehensive understanding of this aspect. To support this addition, we have included an additional sentence in the “Methods” section (Lines 491-492) and appropriate citation in ref. 66, detailing the calculation and visualization performed using the APBS and PDB2PQR plug-in on PyMOL.

Regarding the narrow salt concentration range for salting-in and salting-out, we understand that the reviewer's observation may have been based on comparisons with typical monomeric proteins. For example, ammonium sulfate precipitation, commonly used for protein purification, involves a broad range of salt concentrations to induce protein aggregation or solubilization. In contrast, our system involves active driving forces for assembly, which likely result in the narrower salt concentration range observed in our experiments. Similar ranges of salt effects have been reported for the assembly of actin filament (*Arch. Biochem. Biophys.* 220, 370–378 (1983)) as well as amyloid fibril formation (*J. Biol. Chem.* 294, 15318–15329 (2019); *Biophys. Rev.* 10, 493–502 (2018); *J. Biol. Chem.* 290, 18134–18145 (2015)), which supports the idea that the assembly and disassembly processes in our system are driven by mechanisms akin

to those in such systems. For this reason, we consider the observed salt concentration range in our study to be reasonable and consistent with the nature of the protein assemblies we are investigating.

Comment 6: On structural perspectives, these protein nanotubes are similar to microtubules, but their mechanical properties are closer to microfilaments. The specific reasons need to be discussed and explained.

Response: We appreciate the reviewer's comment on this point. While our design principles were inspired by natural systems, we did not intentionally target the construction of structures mimicking microtubules or actin filaments. Instead, our goal was to establish a design framework that could emulate the flexibility and controllability of assembly seen in natural systems. As a result, we successfully fabricated structures that closely resemble those found in nature, leading us to compare their properties with natural cytoskeletal elements. Through detailed characterization, we identified similarities in certain structural and mechanical properties between our protein nanotubes and microtubules or actin filaments. This motivated us to highlight these comparisons in the manuscript to contextualize our findings within the broader scope of natural protein assemblies.

To address the potential for misleading interpretations, we have revised Line 37 in the manuscript to remove the phrase "reminiscent of the cytoskeleton" as it may have given the impression that we aimed to explicitly replicate cytoskeletal structures. The revised text now accurately reflects the intent and focus of our study.

We thank the reviewer for drawing attention to this point, which allowed us to clarify the objective and implications of our work.

Comment 7: It is worth noting that the flexibility and persistence length of different nanotubes may be different, the results in Fig.3 may only represent the characteristics of one of these nanotubes. In other words, comparisons of persistence length between protein nanotubes and cytoskeletal elements may not be meaningful. Therefore, the relevant conclusions need to be carefully discussed.

Response: We appreciate the reviewer's insightful comment regarding the variability in persistence length among nanotubes with different diameters. As the reviewer correctly points out, our sample contains a mixture of nanotubes with varying diameters, and it is reasonable to assume that each has a different persistence length. To address this point, we estimated the persistence lengths of different nanotubes as follows. Assuming that the monomer is a sphere with radius r , the persistence length L_p^n of a C_n symmetric nanotube is

$$L_p^n = \frac{EI_n}{k_B T}, \quad I_n = \frac{\pi}{4} r^4 \left\{ \frac{2}{\pi^2} (2n)^3 + 2n \right\}, \quad (1)$$

where E is the Young's modulus, k_B is the Boltzmann constant, and T is the temperature (J. Howard, *Mechanics of Motor Proteins and the Cytoskeleton*, 2001, Sinauer Associates, Inc.). Here, we can reasonably assume that the persistence length measured in this study, L_p , is the average of all L_p^n of the C_n symmetric nanotubes observed by cryo-EM, each weighted by its probability p_n :

$$L_p = \sum_{n=4}^{10} p_n L_p^n = E \frac{\pi r^4}{4k_B T} \sum_{n=4}^{10} p_n \left\{ \frac{2}{\pi^2} (2n)^3 + 2n \right\}. \quad (2)$$

Now, r and p_n are obtained from the results of cryo-EM observations (Fig.1b and Fig.3a). Hence, by solving Equation (2), E is calculated as $E = 2.29 \times 10^{-4}$ GPa, and each L_p^n can be calculated using Equation (1). Then, using Equation Line 480 in the manuscript, we obtain the theoretical relationship

between the mean square of the end-to-end distance, $\langle R^2 \rangle$, and the contour length, L , for each C_n symmetric nanotube as follows:

From this graph, we conclude that in the tube length range of this study (shorter than 8 μm), C_4 may be distinguishable from higher symmetry if the tube is long enough, but it is difficult to distinguish the remaining $C_5\sim C_{10}$.

However, this graph also shows that the nanotube is stiffer than intermediate filament (IF), but more flexible than or as flexible as microtubule (MT). Although the difference in the flexibility might be clearer if the nanotube is longer than 15 μm as shown in Supplementary Fig. 7b (in the revised manuscript), we conclude that our results are sufficient to compare the flexibilities between the nanotube and cytoskeletal filaments.

To make sentence more accurate, we have modified a last sentence of section "Diversity and flexibility of tubes" to "Although nanotubes longer than 15 μm would be required for a more accurate estimation of its persistence length, we can conclude that the current results suggest the nanotube is stiffer than intermediate filaments and more flexible than or as flexible as microtubules." (Lines 188–190).

Comment 8: The functions and purposes of introducing D-loop in PuuE-M is illogical and unconvincing, especially considering that considering that the obtained structure is far from the microfilament and that other hydrophobic peptides may also give the same results.

Response: We thank the reviewer for this thought-provoking comment. The suggestion that hydrophobic peptides in general could similarly promote helical structures is highly intriguing and has significant implications for understanding the assembly mechanisms of actin filaments. However, due to the current resolution limitations of our cryo-EM analysis as indicated above (Comment 3&4, Lines 162-164 and Supplementary Movie 1), we were unable to directly test this hypothesis in our study. We recognize this as an important avenue for future investigation.

The introduction of the D-loop in PuuE-M was motivated by observations from persistence length measurements and real-time assembly dynamics, which suggested behavior reminiscent of actin filaments. To explore whether we could emulate actin-like helical structures, we incorporated the D-loop, a region known to be critical for actin filament assembly. As noted in Lines 193 – 198 of the manuscript, the D-loop was incorporated under conditions that closely resemble those in actin protein systems, supporting its relevance and appropriateness for our design.

It is important to emphasize that our goal was not to fully replicate actin filaments but rather to create novel functional structures inspired by actin's design principles. The successful formation of helical

structures in our system demonstrates the feasibility of this approach and validates the functional role of the D-loop in our design.

To address potential misunderstandings (As of comment 6), we have revised Line 37 in the manuscript to remove the phrase "reminiscent of the cytoskeleton," as it may have given the impression that we aimed to explicitly replicate cytoskeletal structures. The revised text now accurately reflects the intent and focus of our study.

We thank the reviewer for drawing attention to this point, which allowed us to clarify the objective and implications of our work.

RESPONSE TO REVIEWER #2

We would like to thank the peer reviewers for their time and effort in co-reviewing our manuscript as part of the Nature Communications initiative. We are also very grateful to the Early Career Researchers and senior reviewers for their valuable comments. Their constructive comments and suggestions were of great help in improving the quality and clarity of our work.

While we cannot specify which specific comments were made by the co-reviewers, we have carefully considered and addressed all points raised in the reports. We are grateful for their thoughtful evaluation and hope that our responses and revisions meet your expectations.

RESPONSE TO REVIEWER #3

We thank the reviewer for their detailed and thoughtful comments, and we appreciate the recognition of the significance of our work, particularly the design and assembly of two-component protein modules into long tubes and the insights into the role of the actin D-loop in assembly mechanisms. The reviewer's acknowledgment of the quality of our cryo-EM analysis and the unexpected role of the D-loop is particularly encouraging.

Comment 1: This is not the first designed tubular assembly, as claimed. The authors should include a reference to Shen et al., Science 2018, from the Baker group, which reported on the design of protein filaments several years ago (DOI: 10.1126/science.aau3775).

Response: Regarding the reviewer's comment about prior examples of designed tubular assemblies, we agree that tubular structures have been reported previously, such as in the work by Shen et al. (Science, 2018) as mentioned and Bethel et al. (Nat. Chem., 2023, as highlighted by Reviewer #4 Comment 10). These studies represent significant advancements in the field, but their designs rely on single-component systems. In contrast, our study demonstrates the assembly of tubular structures using two distinct units, which sets our work apart. The unique aspect of our system lies in the cooperative interaction between two components, which is critical for the formation of the tubular structure. This distinction is why we described our work as "the first" of its kind.

To address this comment, we revised the manuscript to provide a more nuanced discussion of previous studies, clearly acknowledging these important contributions but highlighting the differences in design principles from our study. A dedicated paragraph has been added to the discussion section (Lines 77–80 and Lines 262-266) to clarify this difference and to cite these representative studies (references 18 and 19) appropriately.

We appreciate the reviewer bringing this to our attention and believe that the revised manuscript now provides a more balanced and comprehensive discussion of the context of our work within the broader field.

Comment 2: Apart from that the paper is well-presented using several techniques to analyze the assembly and properties of designed tubes. The authors suggest that the P_{uuE} is positioned on the outside. This could be in principle tested by antibodies against a peptide tag that should be able to access the exposed domains but not protected ones.

Response: We thank the reviewer for their positive comments regarding the presentation of our work and the use of various techniques to analyze the assembly and properties of the designed tubes. We greatly appreciate your suggestion regarding the use of antibodies to distinguish between the inner and outer surfaces of the P_{uuE}-based structures.

We also considered using antibodies for this purpose. However, since both units in our design (P_{uuE}-M and P_{uuE}-p) are derived from the same P_{uuE} protein, it is likely that antibodies generated against these units would bind indiscriminately to both, making it challenging to differentiate between the inner and outer surfaces. For this reason, we focused on detailed structural analysis using cryo-EM. As noted in the manuscript (Lines 162-167), cryo-EM and related analyses suggest that the tubular structures exhibit flexibility and extensibility (Supplementary Movie 1), which further complicates the ability to clearly distinguish the two surfaces at the molecular level.

We would like to note that the recent identification of D-loop-specific antibodies was reported by a separate group (Biochem. J. 481, 1977–1995 (2024)). However, these antibodies are not commercially available and were developed by the authors of that study. Additionally, they have not performed structural analyses to confirm direct binding to the D-loop, making it uncertain whether these antibodies could be reliably used for identifying the inner and outer surfaces of our designed structures.

As an alternative to antibody-based detection, we designed a system in which the D-loop was incorporated into P_{uuE}-p to create P_{uuE}(D-loop)-p and mixed with P_{uuE}-M to test whether tubular structures would form. If the D-loop were positioned internally, the formation of helically coiled structures should be disrupted. Unfortunately, the resulting mixture exhibited poor tube formation, likely due to the internal positioning of the D-loop, which may have hindered proper assembly. While we observed minimal tubular structures, no helical arrangement was detected within them. Given the low yield of tubes, this result does not provide definitive evidence and was therefore not included in the manuscript.

Nevertheless, we believe that the correct assignment of surface orientation in our study is supported by the nature of the D-loop's structural impact. For helically coiled tubes incorporating the D-loop, structural disruption would be difficult to achieve unless the D-loop were positioned externally. This observation lends further confidence to our conclusion regarding the orientation of the surfaces, as presented in the manuscript.

Moving forward, we aim to address this limitation by designing systems based on entirely different protein scaffolds (For primitive data, please see Figure on Reviewer 1 Comment 2). We believe these alternative designs will allow for more detailed analyses, including surface orientation. The results from these ongoing efforts will be reported in future studies to provide a more comprehensive understanding of such assemblies.

We thank the reviewer again for their insightful feedback, which has helped us set clear goals for future investigations.

Comment 3: Additionally, it would be interesting to know the persistence length of the helical assemblies, particularly whether they are longer than the filaments without the D loop.

Response: We appreciate the reviewer's insightful comment. We agree that comparing the persistence length of helical assemblies with and without the D-loop would provide valuable insights. However, the presence of non-helical tubular structures in the sample, even when using units with the transplanted D-loop, complicates such comparisons. As a result, we did not conduct this specific analysis in the current study.

Furthermore, the tubes with the D-loop were typically 1–2 μm in length and did not exhibit significant thermal deformations, making it challenging to estimate their persistence length (L_p) using the conventional analysis employed in this study. Additionally, these results suggest that the persistence lengths of D-loop tubes are much longer than 1 μm , indicating that tubes longer than 15 μm are required to estimate their persistence lengths accurately enough to compare persistence lengths with and without D-loop, as shown in Supplementary Fig. 7b. Therefore, although the reviewer's comment is intriguing, such a comparison remains challenging at this moment.

The formation of larger (longer) assemblies is an essential and promising direction in protein assembly design field, offering significant opportunities for advancing both fundamental understanding and practical applications. Continued efforts in addressing this challenge are expected to contribute to the development of innovative designs and versatile biomaterials with wide-ranging applications.

Comment 4: Also, could they provide information on the interactions between the D loops in the intertwined tubes, is the conformation (at least according to the model) similar to the conformation in actin filaments?

Response: We thank the reviewer for raising this intriguing point. The results suggest that interactions between D-loops, or more generally between hydrophobic interactions, can lead to the formation of helical structures. This finding is not only relevant to our study but also holds significance for understanding the assembly mechanisms of actin filaments.

Unfortunately, the low resolution of the helical structure in the cryo-EM analysis did not allow for a 3D reconstruction that would reveal detailed interactions between the D-loops. This limitation may be due to the flexibility of the tube itself, which hinders the ability to resolve fine structural details. Therefore, we cannot directly compare their conformations to those in actin filaments.

RESPONSE TO REVIEWER #4

We sincerely thank the reviewer for their thoughtful comments and constructive feedback on our manuscript. We are pleased to know that the reviewer found our work interesting and appreciated the clarity and presentation of our design principles, as well as the characterization of the tubular structures.

Comment 1: The acronym used (NIPAD): it is not clear why the methods employed in this study requires this naming scheme as there was no development of new engineering methods or software.

Response: We thank the reviewer for raising this point regarding the use of the acronym "NIPAD." Initially, we used this term to highlight the nature-inspired aspect of our design approach, particularly in contrast to computationally driven methods like those used by the Baker group. However, based on the reviewer's comment, we recognize that the acronym may not clearly reflect the methodologies employed in this study. To address this concern, we have removed the term "NIPAD" from the manuscript and revised the relevant sections accordingly. These changes have been highlighted in the revised manuscript for clarity.

We appreciate the reviewer's suggestion, which allowed us to improve the precision and clarity of our manuscript.

Comment 2: Cryo processing: please add more information, including images of initial models used and how C_1 symmetric reconstructions deviated from the final symmetric reconstructions (if any). This is crucial to understand potential symmetric artifacts at low resolutions, especially for the higher order assemblies.

Response: We thank the reviewer for their constructive suggestion for a better understanding of structural analysis and for proposing methods to validate potential artifacts. This input is valuable for enhancing the robustness of our findings.

Initial models (initial 3D reconstructions) were generated in helical reconstruction using the cryoSPARC software. A featureless hollow cylinder was used as the starting reference volume to avoid reference bias. The inner and outer diameters of this cylinder were approximately measured from 2D class averages.

All 3D classifications in RELION were performed with C_1 symmetry and yielded reconstructions similar to the final reconstruction with imposed cyclic symmetries, thus excluding the possibility of introducing artifacts from cyclic symmetry.

For helical symmetry, we used symmetry search in 3D classification and final reconstructions in RELION to monitor its convergence. Additionally, we generated 2D reprojections from the final 3D reconstructions to examine whether they match the 2D class averages, resulting in a reasonable match.

In response to this comment, we have revised the Supplementary Figures 6, 10 and “Methods” section as follows:

Supplementary Figures 6, 10: Panel labels are attached to the figures, and each step of the image analysis is described in more detail in the legend. New panels have been added showing comparisons between 2D reprojections from 3D reconstructions and 2D class averages.

“Methods” section: A detailed description of the initial model generation and reprojection analyses has been added in section "Cryo-EM image processing (Lines 404-446 as highlighted)".

To support these revisions, we have also included appropriate citation for cryoSPARC in ref. 59.

Comment 3: Methods were not clear as to when binning was reversed during cryoem processing. Was binning reversed for the final resolution calculations?

Response: We thank the reviewer for their question regarding the use of binning during cryo-EM processing. Binning was retained throughout the processing and used for the final reconstructions. From our data analysis, it was evident that the final resolution of the complex did not approach the Nyquist limit with binning. Therefore, we determined that reversing the binning was unnecessary, as all resolvable structural information was preserved in the binned data. We hope this clarifies the methodology employed in our study.

To address this point and avoid confusion, we have added the following description to the “Methods” section (Lines 442-445):

"Particle binning, applied during initial extraction, was maintained throughout the subsequent classifications and 3D reconstructions without reverting to the original pixel size. This approach was adopted as the estimated resolution of the final reconstructions did not reach the Nyquist limit of the binned images, ensuring that all resolvable structural information was preserved."

Comment 4: It is not clear why this design results in tubes and not 2D or 3D crystals or closed c-symmetric oligomers or even large vesicles. Were any observed?

Response: We thank the reviewer for this insightful question. As noted, in addition to tubes, we observed structures suggesting the presence of ring-like assemblies in the negative-stain TEM images as shown in the figure below. Based on our design, as described in the manuscript, the connection site (p66 α) in one of the units (PuuE-p) has a relatively fixed angle (Fig. 1c, Supplementary Fig. 1), which likely promotes the formation of "closed" structures, such as tubes and rings.

We hypothesize that the ring-like structures represent an early stage in the formation of tubular assemblies. For this reason, we focused on analyzing the structures and properties of the tubes in detail. To address the reviewer's comment, we included a zoomed-in image of the ring structures below in Supplementary Fig. 3b, which were derived from the same data as the tubular assemblies.

In addition, we have updated the manuscript to include these observations and explanations, further clarifying the relationship between the ring-like structures and tubular assemblies. (Lines 102-107).

We appreciate the reviewer's comment, which allowed us to highlight this aspect of the observed assemblies.

Comment 5: Looking at the final maps, it seems that additional interactions occur between the scaffold proteins than was originally intended: please discuss.

Response: We thank the reviewer for pointing out the possibility of unintended interactions between the scaffold proteins. While weak interactions between PuuE scaffold units may occur, as suggested, they do not appear to drive the formation of the observed tubular structures. This is supported by the observation that neither PuuE-M nor PuuE-p alone forms higher-order assemblies (Supplementary Fig. 2a, b).

In addition, it is possible that the observed interactions might involve the "Met-6xHis-TEVcs region," which has been excluded for clarity in the figures. As shown in the figure below, the N-terminal of PuuE-p includes the Met-6xHis-TEVcs region, which likely corresponds to the volume enclosed within the framed area in the maps, which we believe corresponds to the "additional interactions" you pointed out.

Taking this into account, we believe it is reasonable to conclude that the critical interactions responsible for the formation of tubular structures arise from the designed interactions between the M3L2 and p66 α components, as intended in our design. If necessary, we would add following sentences after Line 176.

"Furthermore, the 3D maps suggested the presence of weak unintended interactions between PuuE units. However, as no higher-order assemblies containing tubular structures were observed when PuuE-M or PuuE-p were used alone, it is reasonable to conclude that the key interactions driving tube formation are designed interactions between the M3L2 and p66 α components."

We thank the reviewer for bringing up this important aspect, which has helped us further refine the discussion in our manuscript.

Figure. Fitting of the AlphaFold2-predicted model of PuuE-p into the 3D reconstructed model of C_3 PuuE tube as shown in Fig. 4g (left), viewed from a different perspective (right). To improve clarity, the Met-6xHis-TEVcs region of the PuuE-p model is not shown in Fig. 4g. However, the N-terminus of the Met-6xHis-TEVcs region is expected to correspond to the area highlighted in red. We propose that the p66 α of PuuE-p fits into Volume 2, while the N-terminus of the Met-6xHis-TEVcs region aligns with Volume 1, which we believe the reviewer referred to as the "additional interaction."

Comment 6: Figures need greater clarity - for example, figure 1 should more clearly show the interactions that form fibers: it was not easy to understand what the authors see in the design that would create a fiber over other assemblies. Additionally, the symmetric axes should be well defined and repeat units outlined.

Response: We sincerely thank the reviewer for their valuable feedback on improving the clarity of Figure 1. In response to the suggestions, we have implemented the following revisions to enhance the figure and the corresponding explanations in the manuscript:

1. Improved Predicted Model: To address the concern that the design leading to tube formation was not easy to understand, we have revised the predicted model to clearly highlight the interaction sites that contribute to the assembly process (Fig. 1d inlet).
2. Clarification of Angular Connections: To better explain the formation of angular connections leading to closed structures (e.g., tubes), we have updated panel c of Fig. 1. This revision explicitly shows the rigid nature of the p66 α connection site in PuuE-p, as well as the flexible connection in PuuE-M, to clearly illustrate how these features contribute to the resulting assembly.
3. Symmetric Axes and Repeat Units: To address the suggestion regarding symmetric axes and repeat units, we have visually emphasized a repeat unit within a double-framed box (Fig. 1d). Additionally, we have explicitly marked the symmetric axes within the figure to improve clarity. In alignment with this revision, we have also updated Fig. 3b, 4f to reflect the visual emphasis on symmetric axes.
4. Textual Updates: To further support the revised figure, we have fixed explanations in the manuscript (Lines 67-71) to clarify why the design results in closed structures, such as tubes, rather than other assemblies.

These revisions have made our schematic and design approach more transparent and accessible, ensuring that the interactions and structural features are easier to interpret. We greatly appreciate the reviewer's insights, which have allowed us to improve the clarity and presentation of our work. We hope these modifications address your concerns and make the design principles and resulting assembly process more comprehensible.

Comment 7: Cryo processing: did the authors try symmetric expansion and other techniques such as focused refinement to try and discern the interfaces or get higher resolution refinements that better account for dynamics?

Response: We thank the reviewer for this insightful question. We did not attempt symmetric expansion or focused refinement in this study. Given the diversity in tube diameters and symmetries, the seemingly uniform distribution of the energy landscape as shown in Supplementary Movie 1, and the overall low resolution of the final reconstructions, we concluded that these methods would not yield resolutions sufficient to discern individual interface details.

Comment 8: AlphaFold: have the authors tried to understand the assembly using modeling of multiple subunits?

Response: Inspired by the reviewer's comment, we attempted to gain insight into the structure of the tube assembly using AlphaFold2. Since large-scale prediction of the tube assembly was not feasible for our environment, we focused on the smallest repeating unit that makes up the tube.

Initially, we predicted the complex structure using monomers of P_{uuE}-M and P_{uuE}-p. In all five output models, the predicted structures showed interactions primarily between P_{uuE} units, rather than the designed interactions involving M3L2 and p66 α (please see model **a** in figure below).

Next, we modeled tetramers of P_{uuE}-M and P_{uuE}-p to explore potential tetramer-to-tetramer linkages mediated by M3L2-p66 α interactions. To limit M3L2-p66 α interactions to a single pair of subunits, only one set of subunits included M3L2 and p66 α sequences at the C-terminus (P_{uuE}-M and P_{uuE}-p), while the remaining six subunits lacked these sequences (plain P_{uuE}). However, the predicted structures did not display the expected M3L2-p66 α linkage between tetramers. Instead, the models showed either mixed molecules within single assemblies (P_{uuE}-M and P_{uuE}-p in one tetramer) or configurations where molecules were clashed (model **b** in figure below).

To refine this approach, we incorporated a Gly linker strategy, a method commonly used for AlphaFold2 complex structure prediction. Predictions were conducted with linkers of 40 and 50 Gly residues to make tetrameric proteins into a single chain to restrict P_{uuE}-M and P_{uuE}-p separately (model **c** and **d** in figure below). Unfortunately, the resulting structures were like previous predictions (model **b**), displaying domain swapping or molecular clashes rather than meaningful assembly.

Despite these efforts, AlphaFold2 predictions did not yield the structural insights we aimed for regarding the tube assembly. The polypeptide sequences used in the predictions and the highest confidence models (ranked_0) are presented below for reference.

These results suggest that, at present, predicting such higher-order assemblies remains challenging using AlphaFold2. While the tool excels at modeling individual protein structures and small complexes, its ability to accurately capture the interactions and spatial organization required for large-scale assemblies like the tube structure appears to be limited. We believe that future developments in modeling algorithms may provide further insights into these complex assemblies.

Comment 9: Discussion: more detail on the potential applications of these fibers would be beneficial

Response: We thank the reviewer for this suggestion. To address this, we have modified the “Discussion” section to outline potential applications of these fibers (Lines 266–301). These include their use in biomimetic materials, drug delivery systems, creation of artificial cytoskeletons, and the development of artificial transport systems. We hope these modifications provide greater clarity on the broader impact and versatility of our study.

Comment 10: The authors claim that the design of fibers “remains elusive”: fibers have been designed before, such as Bethel et al. 2023 (nature chemistry). Previous fiber studies should be discussed.

Response: We appreciate the reviewer bringing up prior work on fiber design, such as Bethel et al. (Nat. Chem., 2023) and Shen et al. (Science, 2018, as noted by Reviewer #3 Comment 1). These studies indeed represent significant contributions to the field and demonstrate the formation of tubular structures. However, these designs rely on single-unit systems. In contrast, our work involves the cooperative assembly of two distinct units, which sets it apart and contributes to its novelty.

This distinction is the basis for our statement that the design “remains elusive,” as our approach provides a new perspective on controlling assembly through multi-component interactions. To clarify this point, we have added a dedicated paragraph in the discussion section (Lines 77–80 and Lines 262–266), comparing our work with the cited studies and highlighting the unique aspects of our design. We have also included these references in the corresponding section to support our discussion on the properties of the designed structures (references 18 and 19).

Comment 11: Control over assembly: apart from the environmental factors discussed, do the authors have finer control over the fiber assembly and oligomeric states by design?

Response: Thank you for this insightful question. A key highlight of our study lies in the formation of two-component tubular structures, which has not been achieved before, as mentioned in Comment 10. Using these structures, we demonstrated their ability to respond reversibly to external stimuli, achieving controlled assembly and disassembly. The creation of such assemblies that can reversibly respond to external factors remains a significant challenge in protein design, underscoring the importance of our findings.

As the same time, we agree with the reviewer that achieving control over the assembly through intrinsic protein design, rather than external environmental factors, is equally important. Although this aspect extends beyond the scope of the current study, it is an area of great interest to us. We are actively

investigating approaches to achieve such control through new design strategies and hope to present these results in future publications.

Comment 12: D-loop grafted cryoem: the sample was left on ice for 1hr before freezing and fibers were observed, however, this is in contrast to the negative-stain experiments in Fig4c and ExtFig7d, where there are no observable fibers at 0c. Please clarify

Response: We thank the reviewer for this observation. As shown in Fig. 4c (center), Supplementary Fig. 8d (center), and Supplementary Fig. 10 (cryo-EM image), the tubular structure does not disappear completely when the temperature is reduced. Instead, the helical structure is decomposed, but the underlying tubes remain partially intact. This partial preservation of the tubes allows structural analysis even at low temperatures. We hope that this explanation resolves the apparent contradiction. We have addressed this comment and included the explanation in the main text as well (Lines 237-238).

Response to the minor points:

Minor Point 1: "cryo-electron": the hyphen should not be between cryo and electron.

Response: We thank the reviewer for pointing this out. The manuscript has been revised to correct this terminology (Lines 17 and 156-157).

Minor Point 2:"Figure 4g: number of atoms outside contour is not very useful without knowing the threshold used and where the greatest deviation between map and model lies."

Response: We agree with the reviewer that additional clarity is needed. To address this, we changed the words with "visual inspection" that more effectively communicate the discrepancies between the map and model. These changes were implemented in the revised Figure 4g.

Minor Point 3: "The role of M3L2/p66a in the MBD2-NuRD complex is not described."

Response: We thank the reviewer for pointing out this aspect. Upon review, we have determined that MBD2 is not directly relevant to the scope of this study. Therefore, we have revised the manuscript to remove references to MBD2, focusing solely on the role of M3L2/p66a in the NuRD complex (Lines 49-52). This adjustment ensures clarity and maintains the relevance of the discussion to the current study. Accordingly, we have changed ref. 11 to more appropriate citation.

We greatly appreciate the time and effort that the reviewers have invested in evaluating our work and providing insightful comments that have helped us refine our study.

We were encouraged by the reviewers' recognition of our study's novelty and potential impact, particularly in the design of two-component protein assemblies. At the same time, we carefully considered the additional comments and suggestions provided in the most recent round of review and have revised our manuscript accordingly. In this revised version, we have addressed the remaining concerns raised by the reviewers through further clarifications in the text and additional discussions where necessary.

RESPONSE TO REVIEWER #1

Comment: The reviewer's comments and concerns have been adequately addressed. Although the additional experiments involving cryo-EM and SAXS to investigate the formation mechanisms were not performed, the dynamic and responsive properties demonstrated in this study represent a significant advancement in the field of protein assembly. This is supported by recent reports highlighting the importance of such properties (e.g., *Nat. Nanotechnol.* 2024, 19, 1016; *Nat. Chem. Biol.*, 2025, doi.org/10.1038/s41589-024-01811-1; *Nat. Synth.*, 2025, doi.org/10.1038/s44160-024-00726-y; *Chem*, 2025, doi.org/10.1016/j.chempr.2024.102407). In its current form, this manuscript meets the standards for publication in *Nature Communications*.

Response: We sincerely appreciate the reviewer's positive evaluation and thoughtful comments on our manuscript. We agree that understanding the formation mechanisms of these assemblies, as well as developing precise control over their structures, represents a crucial step in the broader study of protein assembly. Building upon the findings of this study, we aim to further explore and refine the design of these structures to advance the field.

Thank you again for your insightful feedback and for recognizing the significance of our work.

RESPONSE TO REVIEWER #2

Comment: I co-reviewed this manuscript with one of the reviewers who provided the listed reports. This is part of the *Nature Communications* initiative to facilitate training in peer review and to provide appropriate recognition for Early Career Researchers who co-review manuscripts.

Response: We deeply appreciate your efforts in co-reviewing our manuscript as part of the *Nature Communications* initiative. We are grateful for the constructive feedback provided, which has greatly contributed to improving our study.

Thank you again for your time and thoughtful evaluation.

RESPONSE TO REVIEWER #3

Comment 1: Authors have mainly modified the text of the manuscript without additional data or insights. Although there are still some unanswered questions (e.g. regarding the role of the “D-loop”) and challenges, the manuscript nevertheless describes an interesting design of protein fibers composed of two components. The authors built their assembly based on natural homotetramers and modified them by weak heterodimeric domains to make a two-component assembly. This approach could be applied in principle to many other building blocks and therefore represents a new design platform.

Response: We appreciate the reviewer’s recognition of our approach as a potential new design platform applicable to other building blocks. This encouragement reinforces our belief that our strategy can contribute to expanding the field of artificial protein assemblies.

In this study, we took a novel approach by incorporating the D-loop—a key structural element essential for the helical organization of actin filaments—into an artificial protein assembly. While previous studies have reported actin mutants with modified D-loops using expression systems such as insect cells, it remains difficult to isolate the functional contribution of the D-loop alone in the native actin context due to the complexity of its overall structure and interactions.

Our aim in this study was to demonstrate a new bottom-up approach in protein assembly design, wherein a naturally important structural motif is integrated into an artificially designed protein assembly with a similar morphology—in this case, a tubular structure—to reveal its function in a simplified and controllable context.

By incorporating the D-loop into a structurally unrelated scaffold protein, we created a modular system that allows for the evaluation of this key motif’s contribution independent of the native actin framework. We believe that this study successfully demonstrates a proof of concept, highlighting a new application of artificial protein assemblies for dissecting functional elements derived from natural proteins. Furthermore, as suggested in Reviewer #1’s Comment 8, we plan to conduct further experiments in the future to achieve a more detailed characterization of the role of the D-loop in our system.

To further clarify the conceptual motivation behind incorporating the D-loop into our design, we have added new sentences in the Discussion section of the revised manuscript (Lines 282–291). This addition emphasizes our intention to demonstrate a novel usage of structurally important motifs by integrating them into artificial scaffolds, rather than conducting a detailed functional analysis of the D-loop itself. We hope this helps to better convey the design philosophy underlying our study.

Comment 2: While authors did not report success by modeling the assembly using AF2, I Reviewer #3) obtained formation of 4+4 assembly linked through the CC peptides using AF3 algorithm (image of the models added). 4+4 assembly represents a dead end as it is not able to form larger filaments but may represent an intermediate in equilibrium with filamentous assemblies. Authors may invest a bit more effort into modeling and explanation of the putative mechanisms.

Response: We sincerely appreciate the reviewer’s efforts in applying the AF3 algorithm and sharing their findings. We agree that the 4+4 assembly could be considered one of the initial intermediates. However, given its structural and energetic disadvantages, it is likely to transition into a more stable configuration over time.

While experimentally distinguishing these intermediates remains technically challenging at this stage, we have carefully examined available data and observations to support our conclusions. We acknowledge the significance of this question and consider it an important direction for future studies as experimental techniques advance.

Thank you again for your insightful comments, which have helped us refine our perspective on the assembly process.

Comment 3: One issue that might be discussed is whether the two components need to be produced separately before mixing, to avoid formation of heterotetramers, which might hinder the formation of filaments. Did they ever try the production of both components at the same time?

Response: We appreciate the reviewer’s insightful question. In this study, we did not attempt the simultaneous production of both components. This decision was based on the consideration that the parental protein (PuuE) forms a tetramer upon expression in *E. coli*, which could lead to the random mixing of PuuE-M and PuuE-p at the monomer stage (as illustrated below). Such heterotetramer formation would likely hinder the formation of the well-defined structures observed in our study. Currently, we are exploring this approach using a different scaffold protein system as shown in previous Point-by-point response to the reviewers’ comments (Reviewer #1 Comment 2) , and we aim to publish these findings in a future study.

Additionally, producing the two components separately provides better control over the structural formation, as demonstrated in previous studies (e.g., *Nature* 589, 468–473, 2021). Based on these considerations, we believe that this strategy is the most suitable for achieving the intended assembly and structural control, and therefore adopted the approach of separately producing and subsequently mixing the two components in the present study.

Comment 4: As the authors concur that they did not demonstrate any cytoskeletal function and removed the phrase “reminiscent of the cytoskeleton”, they might as well remove the reference for the cytoskeleton from the title.

Response: We appreciate the reviewer’s suggestion. Our study was not specifically designed to mimic the cytoskeleton but rather aimed to construct well-defined protein assemblies with controllable structures. As a result, we successfully formed structures that exhibit similarities to cytoskeletal assemblies. Given this outcome, we believe that retaining the reference to the cytoskeleton in the title is appropriate.

We hope this clarification addresses the concern.

RESPONSE TO REVIEWER #4

Comment 1: Thank you for your efforts in addressing my concerns. Most concerns have been addressed, however, I have follow up comments below:

Response: Thank you for your careful evaluation and for taking the time to provide additional feedback. We appreciate your thoughtful comments and address them in detail below.

Comment 2 (Regarding Comment 2 from previous Point-by-point response to the reviewers’ comments): You have not included images of the initial models used, only a description. Including the initial models will help compare to the final reconstruction.

Response: We appreciate the reviewer’s valuable feedback. We agree that this addition would enhance the clarity and completeness of our study, allowing readers to better visualize the progression from initial models to final reconstructions.

In response to the comment, we have added panels (Supplementary Fig. 6 and 10) showing representative images of the initial models used for each structure described in the study.

Comment 3 (Regarding Comment 5 from previous Point-by-point response to the reviewers' comments): I think your text is reasonable to include, however, you should actually check how much your tag is influencing the formation or stability of the tubes by cutting the protein using tev. The construct already has this feature.

Response: We appreciate the reviewer's suggestion. We added the following sentences after Line 176.

"Furthermore, the 3D maps suggested the presence of weak unintended interactions between PuumE units. However, as no higher-order assemblies containing tubular structures were observed when PuumE-M or PuumE-p were used alone, it is reasonable to conclude that the key interactions driving tube formation are designed interactions between the M3L2 and p66 α components."

As the reviewer suggested, we initially attempted to remove the tag using TEV protease after protein purification. We also considered that removing the tag would be preferable from an experimental standpoint, as it would allow us to more directly evaluate the intrinsic assembly behavior of the designed proteins. However, during the treatment and subsequent purification steps, we observed a significant loss of protein, which made it technically impractical to proceed with tag removal under our current conditions. Therefore, we carried out the experiments using the tagged proteins.

Despite the presence of the tag, the designed higher-order structures were successfully formed, supporting the validity of our approach. Additionally, the possibility of 6xHis-tag-induced aggregation can be ruled out, as EDTA was included in the buffer to chelate metal ions and prevent such interactions. While other potential interactions involving the tag cannot be entirely excluded, no higher-order assembly was observed when each component was incubated separately (Supplementary Fig. 2a, b). Based on these observations, we conclude that the tag does not contribute to the formation of the assembled structures.

We have carefully considered the potential influence of the tag in our study and reached the conclusions presented in this manuscript. Since this evaluation did not involve new experimental data or lead to substantial changes in interpretation, we have chosen not to include this discussion in the main text, but instead provide our rationale here in response to the reviewer's suggestion. We appreciate the reviewer's valuable suggestion and hope that our explanation clarifies our rationale.

Comment 4 (Regarding Comment 7 from previous Point-by-point response to the reviewers' comments): I don't think I fully agree. While you may not reach high resolution and for some of your reconstructions I agree, you may not get more information, for others you could get more resolution leading to enough resolution to discern helices (such as your 9.7Å reconstruction of the c3 tube). Your interface would be revealed to greater detail and would help validate your design.

Response: We appreciate the reviewer's thoughtful comment regarding the potential for improved resolution in our reconstructions, particularly for the C₃ tube.

In light of your suggestion, we have revisited our reconstruction process and obtained a reconstruction using boxes with the original pixel size. This yielded a reconstruction similar to those presented in the manuscript, with comparable helical parameters and a resolution of around 10 Å. This outcome aligns with the reconstruction using the downscaled boxes.

Nonetheless, it would be beneficial to explain these efforts in the manuscript. We have included a brief explanation detailing our attempts to optimize resolution, including using boxes with the original pixel size (Lines 455–461). This addition will give readers a more comprehensive understanding of our processing pipeline and the steps taken to maximize the information obtained from our data.

Comment 5 (Regarding Minor point 2 from previous Point-by-point response to the reviewers' comments): Again, this is not effective without knowing the threshold used. Different values will "contain" different amounts of atoms. The new term of "visual inspection" is not good to use either way.

Response: We appreciate the comment regarding the thresholds used for map visualisation. We agree that our current description using "visual inspection" is not sufficiently precise or reproducible.

To address this concern, we have replaced the term "visual inspection" with a more quantitative description of our thresholding approach. Specifically, we now refer to this process as "structure fitting" in Fig. 4g, which is based on a normalised map threshold ($\sigma = 3.3$). In accordance with this change, we have also replaced the previously used map with a normalised version to ensure consistency with the sigma-based thresholding. Furthermore, we re-evaluated the number of atoms outside the density at this threshold and updated the corresponding values. We have explicitly stated the sigma value used for map visualisation in the figure legend to ensure clarity and reproducibility. Accordingly, we have included the following sentences in the legend of Fig. 4g (Lines 721–723):

"The map density was normalised and visualised at a threshold of sigma (σ) value = 3.3 above average. Structure fitting was performed to count the atoms of PuuE-p model outside the reconstructed 3D map of C₃ tube structure at this threshold."

This approach will allow for a more objective assessment of the density distributions in our maps. We sincerely appreciate the reviewer's insightful feedback, which has helped us refine our terminology and adopt a more scientifically precise and unambiguous description.

RESPONSE TO REVIEWER #3:

Comment: I find the manuscript appropriate for publication due to its novelty and potential interest. I still believe that the assembly they produced is not related to the cytoskeleton. Still, I don't insist on removing the word from the title, where its function seems to be mainly to attract the attention of the readers.

Response: We sincerely appreciate the reviewer's positive evaluation regarding the novelty and potential interest of our study.

Regarding the reference to the cytoskeleton in the title, we fully understand the reviewer's concern. As discussed in our previous response, while our study was not specifically designed to mimic cytoskeletal function, the assemblies we created exhibit morphological similarities to cytoskeletal structures. Our intention in referencing the cytoskeleton was not to claim functional equivalence, but rather to highlight the structural resemblance, which may inspire further exploration of cytoskeleton-like dynamics in synthetic systems.

Given this context and considering that the reviewer has kindly indicated no strong objection to retaining the term, we respectfully propose to maintain the current title. We sincerely appreciate the reviewer's understanding and flexibility on this point.

RESPONSE TO REVIEWER #4:

Comment: Thank you for including extra information. I do have the following comments for the Authors, which are mainly comments from a previous review that the Author's have not fully addressed yet.

I think clearing the following can be done and given that, I think this manuscript would be ready for publication.

Response to overall comment: We sincerely appreciate the reviewer's careful re-evaluation and constructive comments. We are encouraged by the reviewer's recognition that only a few clarifications remain before publication. In the following responses, we have addressed each of the remaining points in detail and made clarifications where necessary.

Comment 1-1: My previous comment of "You have not included images of the initial models used, only a description. Including the initial models will help compare to the final reconstruction".

Thank you for adding these, but they contradict your methods section where you state that the initial models were hollow cylinders as these initial models are not generic hollow cylinders. Can you please clarify and update the text and/or figures appropriately?

Response: We appreciate the reviewer's careful reading and for pointing out the apparent inconsistency between our methods section and the newly added images. We apologize for any confusion this may have caused. To clarify, a featureless cylinder was used as the initial reference for the first iteration of "Helical Refinement" in cryoSPARC, but this was not clearly addressed in our original description. We have now updated the Methods section (Lines 427–432) as follows:

“Initial 3D reconstructions were carried out using the "Helical Refinement" algorithm in cryoSPARC. A featureless hollow cylinder was used as the starting reference for the first iteration in refinement to avoid reference bias, with inner and outer diameters approximately measured from

2D class averages. The resulting helical structure from this initial refinement job was then used as the reference volumes for subsequent analyses in RELION.”

We hope this clarification addresses your concern and provides a more accurate description of our refinement process.

Comment 1-2: You don't mention if the models were low pass filtered for refinement. Please address.

Response: Thank you for bringing this important point to our attention. We did indeed apply low-pass filtering during the 3D classification and refinement process to avoid reference bias. We have updated the Methods section to include this important detail (Lines 456–458) as follows:

"For 3D classification and refinement, the initial volumes were low-pass filtered to 40 Å resolution to avoid reference-based artifacts.”

We hope this revision clarifies our methodology.

Comment 2: My previous comment "I think your text is reasonable to include, however, you should actually check how much your tag is influencing the formation or stability of the tubes by cutting the protein using tev. The construct already has this feature"

I can understand your reasoning for keeping the tag, however, do you see any formation of the fibers after tev cleavage? Even with low concentrations, you can easily validate this using EM images and rule out of the effect of the tag.

Response: Thank you very much for your valuable comment and suggestion regarding validation after TEV cleavage.

To address this point, we performed additional experiments using commercially available, highly active TEV protease, as our previous in-house preparation had lower efficiency. After treating the purified proteins with the fresh TEV protease, we observed that the overall protein recovery was improved compared to our earlier attempts. We were able to obtain a sufficient amount of protein to conduct testing experiments, although the protein concentration used for these tests was kept relatively low (6.25 μM).

We then tested the ability of the cleaved samples to form tubes. Under the standard conditions described in our manuscript, we did not observe tube formation with the cleaved proteins. However, when we increased the NaCl concentration to 400 mM, tubular structures were clearly observed by TEM as shown below.

200 nm

200 nm

We hypothesize that this behaviour is due to a decrease in the isoelectric point (pI) of the proteins after removal of the His-tags, resulting in a higher requirement for ionic screening to promote assembly. To support this hypothesis, we calculated the theoretical pI values before and after TEV cleavage: 6.44 to 6.38 for PuuE-M, and 6.02 to 5.84 for PuuE-p. Given that tetramerization would be expected not to significantly alter the overall pI, these results suggest that the cleaved proteins indeed require higher ionic strength to promote assembly due to enhanced net negative charge. This interpretation is consistent with the mechanism proposed in our manuscript, where modulation of electrostatic interactions plays a key role in controlling assembly behaviour.

We note that under the 400 mM NaCl condition, some aggregation was also observed, as was previously seen with the His-tagged proteins. This suggests that at this higher ionic strength, nonspecific aggregation may be promoted alongside tube formation. Nevertheless, clear tubular structures were distinguishable by TEM, and the observed behaviour remains consistent with our proposed assembly mechanism.

Since these findings do not change the main conclusions of the study and were obtained under slightly modified experimental conditions, we have not made additional modifications to the main text. We sincerely appreciate the reviewer's insightful suggestion, which helped further validate the robustness of our assembly design.

Comment 3: My previous comment of "I don't think I fully agree. While you may not reach high resolution and for some of your reconstructions I agree, you may not get more information, for others you could get more resolution leading to enough resolution to discern helices (such as your 9.7Å reconstruction of the c3 tube). Your interface would be revealed to greater detail and would help validate your design."

The author's did not address this point as they mixed up my suggestion with another suggestion regarding binning and using the full pixel size. My comment was regarding using symmetry expansion or masking/local refinements to potentially improve the resolution of some of your reconstructions, mainly the one at 9.7Å. This can be done using the binned particles. I think this could improve the resolution potentially to the 6-8 range where helices would be better defined and better validate your designed interface.

Response: Thank you very much for your thoughtful and constructive suggestion regarding the use of symmetry expansion and masking/local refinements to improve our reconstructions, particularly at the intermolecular interfaces. We sincerely apologize for the confusion in our previous response, where we mistakenly addressed a different comment related to pixel size and binning.

Following your recommendation, we performed symmetry expansion as well as local refinements with masking using cryoSPARC on the unbinned particles. However, these attempts did not result in notable improvements in either the overall resolution or the map interpretability.

As demonstrated in Supplementary Movie 1, we suspect that the lack of improvement stems from the intrinsic flexibility between the base domain and the connection domain of our fusion protein. These domains are connected via a relatively flexible linker, which allows them to adopt independent conformations. This dynamic behaviour likely prevents the protein from being treated as a rigid single particle during reconstruction, thus limiting the effectiveness of focused refinements.

Since these additional refinements did not yield improved resolution, and the current discussion already reflects the limitations imposed by structural flexibility, we have not made further modifications to the main text in response to this comment. Nonetheless, your suggestion greatly contributed to our deeper understanding of the conformational heterogeneity within our assemblies, and we are sincerely grateful for the opportunity to explore this important aspect in greater detail.

RESPONSE TO REVIEWER #4:

Comment: Thank you for looking fully into my questions. I only have additional comments in regard to "Comment 2", tev cleavage of his-tag.

The aim of this experiment was to check that the tubular assembly is not influenced by the his-tag and based on your results, it seems to be. Based on this, I suggest the authors discuss this in the manuscript. Are the assemblies consistent in size and morphology to pre-cleaved assemblies?

Response: In response to the final comment from Reviewer #4 regarding the potential influence of the His-tag on tubular assembly, we have now addressed this point directly in the revised manuscript (Lines 142–152) and added Supplementary Figure 5c. Additionally, we have revised the legend of Supplementary Figure 5 to include the following description for panel c:

c, 6.25 μM of His-tag cleaved PuuE-M and His-tag cleaved PuuE-p each in 400 mM NaCl buffer was incubated at 40 °C for 24 h and imaged via nsTEM.

We clarified that while the TEV-cleaved proteins were still capable of forming tubes with comparable morphology, the process required a significantly higher NaCl concentration. This observation suggests that the surface charge modulation introduced by the His-tags does influence the assembly behavior under standard conditions. Accordingly, we have updated the text to reflect this interpretation and provided a rationale for retaining the His-tag in subsequent experiments. Details of the TEV protease treatment protocol have also been added to the Methods section (Lines 410–418).

[Comments of Reviewer #1 are below.]

In this manuscript, Suzuki et al. reported an artificial protein assembly system that exhibits reversible self-assembly behaviours under biomimetic conditions. The assembly structures were characterized by cryo-EM, and the mechanical properties and conditional responses of protein nanotubes were investigated using TIRFM, nsTEM and CD. However, after thoroughly reading, it was found that the main ideas of this paper lack novelty, and the authors did not explore the mechanisms of many observations in depth and show potential applications, so this paper may not be suitable for publication in Nature Communications.

1. The main concept of Nature-Inspired Protein Assembly Design is ambiguous and unconvincing, as many works have established reversible protein nanotubes that respond to biomimetic conditions (Nat. Commun. 2022, 13, 5424.; J. Am. Chem. Soc. 2018, 140, 1, 26.; Nat. Chem. 2013, 5, 613.; J. Am. Chem. Soc. 2013, 135, 31, 11509.; ACS Catal. 2020, 10, 9735.). Also, in this work, a meaningful application of these nanotubes have yet to be demonstrated and explored.
2. The definition of two-component tube structures is controversial. Here, the main scaffold of P_{uu}E-M and P_{uu}E-p is the same, but the two-component protein assembly should be two independent ingredients (Nature 2014,510, 103.; Science 2016, 353, 389.; Nature 2021, 589, 295 468-473).
3. Although the structures of protein nanotubes are supported by 3D reconstruction, the underlying mechanisms of the formation of these structures have not yet been explored and explained. Why do different nanotubes emerge under the same assembly conditions? Could certain conditions generate dominant species?
4. Tracking the formation of nanotubes under thermally stable conditions is important to gain insight into the self-assembly mechanism using cryo-EM, SAXS, or other characterization methods.
5. The overall electrostatic potential of P_{uu}E-M and P_{uu}E-p should be shown in Extended Data Fig. 5, especially since the salt concentration range for salting-in and salting-out is very narrow compared to most proteins.
6. On structural perspectives, these protein nanotubes are similar to microtubules, but their mechanical properties are closer to microfilaments. The specific reasons need to be discussed and explained.
7. It is worth noting that the flexibility and persistence length of different nanotubes may be different, the results in Fig.3 may only represent the characteristics of one of these nanotubes. In other words, comparisons of persistence length between protein nanotubes and cytoskeletal elements may not be meaningful. Therefore, the relevant conclusions need to be carefully discussed.
8. The functions and purposes of introducing D-loop in P_{uu}E-M is illogical and unconvincing, especially considering that considering that the obtained structure is far from the microfilament and that other hydrophobic peptides may also give the same results.